# Cost increase in the electricity supply to achieve carbon neutrality in China

Zhenyu Zhuo [1], Ershun Du[2], Ning Zhang [1✉], Chris P. Nielsen [3], Xi Lu [4,5], Jinyu Xiao[6], Jiawei Wu[6] & Chongqing Kang [1✉]

The Chinese government has set long-term carbon neutrality and renewable energy (RE) development goals for the power sector. Despite a precipitous decline in the costs of RE technologies, the external costs of renewable intermittency and the massive investments in new RE capacities would increase electricity costs. Here, we develop a power system expansion model to comprehensively evaluate changes in the electricity supply costs over a 30-year transition to carbon neutrality. RE supply curves, operating security constraints, and the characteristics of various generation units are modelled in detail to assess the cost variations accurately. According to our results, approximately 5.8 TW of wind and solar photovoltaic capacity would be required to achieve carbon neutrality in the power system by 2050. The electricity supply costs would increase by 9.6 CNY¢/kWh. The major cost shift would result from the substantial investments in RE capacities, flexible generation resources, and network expansion.

[1] Department of Electrical Engineering, Tsinghua University, Beijing 100084, China. [2] Institute of Climate Change and Sustainable Development, Tsinghua University, Beijing 100084, China. [3] Harvard-China Project on Energy, Economy and Environment, John A. Paulson School of Engineering and Applied Sciences, Harvard University, Cambridge, MA 02138, USA. [4] School of Environment, Tsinghua University, Beijing 100084, China. [5] Institute for Carbon Neutrality, Tsinghua University, Beijing 100084, China. [6] Global Energy Interconnection Development and Cooperation Organization, Beijing 100031, China. ✉email: ningzhang@tsinghua.edu.cn; cqkang@tsinghua.edu.cn

During the 75th UN General Assembly, the Chinese government announced that China aims to reach peak carbon emissions before 2030 and achieve carbon neutrality by 2060 (the so-called "3060" targets)[1]. The electricity sector in China consumes over 50% of the total coal supply nationally, leading to annual carbon dioxide ($CO_2$) emissions of 4.5 billion tonnes[2]. Numerous studies have indicated that the power sector must take the lead among China's sectors by achieving national carbon neutrality by 2050, in advance of the economy-wide target of 2060[3]. In 2019, 62.1% of China's electricity supply came from coal-fired units. Although China already features the world's largest installed power generation capacity for renewable energy (RE), a profound transformation of the power system will still be required over the next 30 years to achieve carbon emission goals.

The total unit capacity of the key technologies required for decarbonization, e.g., variable renewable energy (VRE, defined here as wind and solar PV power), increased by a factor of ten in the past decade, due to massive feed-in tariffs and other policy support[4]. The VRE capital cost per kW has decreased significantly, with values for PV and wind power decreasing by 73.3% and 15.9%, respectively, since 2012 in China[2,5]. The declining trends in the cost of RE technology are clearly helping to decarbonize the power system[6,7].

However, the integration of VRE entails considerable external costs to guarantee secure and stable power system operations. The intermittency and uncertainty of VRE outputs require the coordinated dispatching of flexible generation resources, such as conventional units or energy storage systems. Moreover, the significant geographic mismatch between load centres and RE generation centres requires extensive grid interconnection. As the installed capacity increases, the completion of projects in areas with the best wind and solar resources will force later development into lower-quality regions, thus increasing the levelized cost of energy (LCOE). In addition, the need to achieve carbon neutrality in a short time frame of 30 years may force development that strains manufacturing capacities and induces massive investment in RE units before their cost decline has been fully realized[8]. Such costs of the transition to carbon neutrality are affected by many factors, and the scale of these effects remains unclear. The costs of the power system transition are eventually paid by consumers through renewable surcharges in electricity bills. It is, therefore, crucial to evaluate the factors driving the electricity supply cost for carbon neutrality to inform effective energy policy design and implementation.

Previous studies have explored the wind and solar potential in China and other regions around the world[9-11], focusing on the natural resource endowments. Based on these results, RE capital costs considering the external costs of RE integration are studied at a fine spatial resolution[12]. To assess the impacts of low-carbon goals on power systems, studies on the resulting power system morphology[6,13,14], comparative development status across countries[15] and associated environmental benefits[16,17] have been conducted. However, few studies have focused on the costs of the electricity supply required to achieve carbon neutrality, including detailed consideration of the practical challenges associated with high renewable capacity shares. Simply estimating changes in electricity supply costs based on only the power balance will result in the underestimation of the comprehensive effects of critical factors. In particular, thorough cost accounting must consider the VRE supply curves, the operating security concerns caused by the high uncertainty and low inertia of VRE units, and the projection of interprovincial network developments.

To accomplish these tasks, we established a power system planning model with a high RE capacity share to evaluate the impacts on the electricity supply cost associated with China's transition to carbon neutrality. The power system generation and transmission expansion is formulated, and the changes in power system morphology over time are simulated. The stability problems caused by high RE penetration, such as decreased inertia and lack of peak regulation reserves, are considered in detail via security constraints in the planning model. The supply curves of VRE based on high-resolution meteorological and geographic data are integrated into the planning model to comprehensively assess costs. Furthermore, a simulation of the long-term development of China's power system at the province level is conducted. For the local grids including within-provincial transmission networks and distribution networks, we project expansion based on historical network data and simulation results. Finally, the modelling results provide an accurate estimation of the external costs of RE integration into the overall power system.

## Results

**Model and scenarios.** The electricity supply cost changes caused by the carbon neutrality transformation are evaluated based on the power system expansion planning model considering the RE distribution characteristics and system security constraints under high renewable penetration. The model structure is presented in Fig. 1. The model simulates generation and transmission expansion from 2020 to 2050 while considering environmental constraints and energy policy goals at the macro-level. The operation security and stability constraints at the micro-level are also considered. The modelling formulation, assumptions, and parameters are presented in the "Methods" section and Supplementary Note 1.

China's wind and solar PV potential and supply curves in this paper are assessed using the Global Renewable-energy Exploitation Analysis (GREAN) database. The GREAN database contains detailed geographic information, RE resource endowment data, and human activity information for the quantitative evaluation of economic potential. Figure 2 presents the spatial distributions of VRE capacity factors and regional VRE supply curves across mainland China. The total economic potentials for wind and solar PV are 7.2 TW and 128.1 TW, respectively. The total energy potential is 200.9 PWh per year, which is 13.5 times China's maximum projected electricity demand in 2050 (see "Methods" and Supplementary Note 5 for more details on the evaluation methods and VRE potential).

Although the total potential is high, the spatial distribution and corresponding LCOEs are not uniform and are highly mismatched with the electricity demand. RE-rich areas for both wind and PV are mainly located in northern and northwestern China, while the load centres are in coastal areas. The VRE supply curves for each province are detailed in Supplementary Figs. 7 and 8. With the accelerated development of RE, the LCOE increases remarkably due to the decrease in capacity factors and the increasing difficulty of construction and grid integration due to the growing distance from existing transmission lines and other factors. This trend is particularly obvious for wind power. The cost distribution for onshore wind displays a "fat tail" pattern, which means that the costs are distributed over a large range. The LCOE ranges from 17.5 to 54.8 CNY¢/kWh in different regions (95% confidence level), with a nationwide average of 26.7 CNY¢/kWh. The LCOE distribution for PV is relatively concentrated, and varies from 11.7 to 31.5 CNY¢/kWh (95% confidence level), with a nationwide average of 19.5 CNY¢/kWh. The RE potential and supply curves in each province determine the development sequence and spatial distribution of renewable energy units, thus influencing the future power system morphology under carbon neutrality targets. Therefore, it is necessary to consider the supply curves across the whole country when optimizing RE investments and regional network connections.

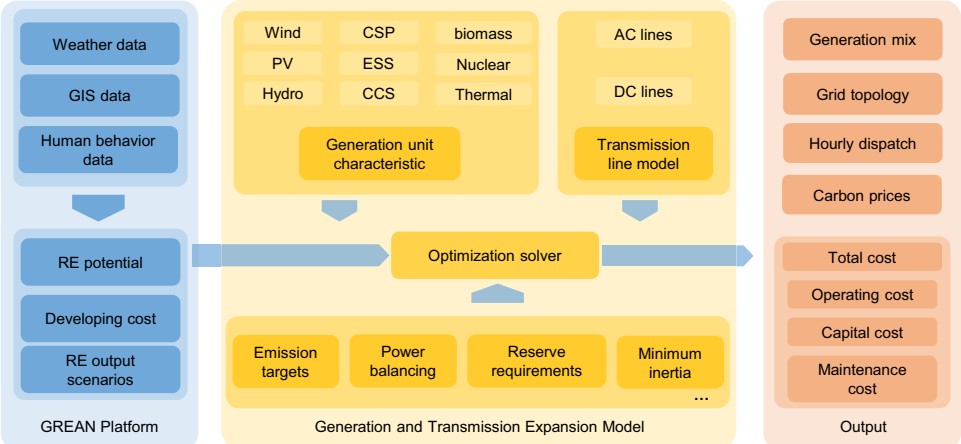

**Fig. 1 Modelling framework for generation and transmission expansion optimization.** We generate the typical year-round hourly output data for renewable energy (RE) plants and summarize the supply curves for each province based on the Global Renewable-energy Exploitation Analysis (GREAN) database. The generation and transmission expansion (GTEP) model, an optimization problem solver, simulates future power system developments. The hourly RE output intermittency, the characteristics of various generation technologies, and system security requirements are all considered. The impacts of high RE penetration are analyzed based on the solution results.

To precisely calculate the electricity supply costs of a power system with high renewable penetration, we have considered the following critical factors and new features in detail:

- The VRE supply curves for each province are integrated into the GTEP model in a piecewise manner (see Supplementary Note 2 for details). In addition to VRE-related data, the future development conditions and potential for other generation resources in China such as hydropower, biomass energy, and natural gas are explored based on regulatory reports and expert consultations.
- The model includes three kinds of power system security and stability constraints: power reserve limits, spinning reserve requirements, and minimum system inertia limits. The three constraints correspond to the security challenges of high-RE-penetration power systems in terms of planning, operation, and transient stability. Such constraints determine the additional flexible resources that are required to accommodate the increasing RE.
- We set potential candidate transmission lines for nation-wide grid connections in the future considering the locations of generation and loads. The model also accounts for the capital costs needed in provincial grids. The local network expansion including within-provincial transmission and distribution networks is projected according to GTEP simulation results and historical network data in an ex-post fashion (see the "Methods" section).
- The capacity mix and the interprovincial network topology of China's power system as of 2020 are considered in the model. Other key parameters such as equipment costs and load demands are also updated to 2020. The parameters of the generation and transmission technologies are presented in Supplementary Tables 14–20.

Such detailed modelling of RE development costs, power system security constraints, and grid expansion investments ensure an accurate assessment of electricity supply costs with the carbon neutrality target. In addition, compared with models in the existing literature, our model provides advantages related to power equipment modelling, a high temporal resolution, and a long planning period duration. The details are discussed in Supplementary Note 6. Other settings and assumptions on the GTEP models are presented in Supplementary Note 1.1.

Four carbon mitigation scenarios considering different government policies are designed to compare electricity supply costs with increased carbon mitigation. The specific descriptions for the scenarios are shown in Table 1. The carbon emission trajectories and load demands of the four scenarios are presented in Fig. 3. The load demands met by the power sector differ among scenarios because the electrification levels of other industries also differ by scenario. The power sector needs to bear a larger electrification load under more stringent carbon emission goals. The annual load demands at the provincial level in each of the scenarios are presented in Supplementary Tables 21–23.

Note that the "3060" target for China applies to the entire economy and society. Carbon neutrality is easier to achieve in the power sector than that in other sectors due to the relative maturity of low-carbon generation technologies. In addition, the power sector must achieve carbon neutrality in advance of the manufacturing sector, the transportation sector and other sectors, to provide zero-carbon power for the anticipated load increase from electrification. Hence, we regard the CN2050 scenario, which achieves carbon neutrality for the power sector in 2050, as a practical translation of the "3060" target to the power sector, and we set it as the base case for comparison and analysis. The electricity supply costs and the variation in the power system morphology, including the capacity mix and transmission network structure, are calculated for each scenario.

**Power system morphology under carbon neutrality.** Figure 4 presents the capacity mix and generation mix under each scenario in 2050 according to our GTEP model. For the carbon-neutral target, the total installed capacity in 2050 will be 3.7 times the current capacity. Regarding the capacity mix, the carbon emission reduction targets mainly impact the choice between coal power and RE. In BAU without emission limits, the coal-power capacity increases from 1104.5 to 1409.4 GW in 2050, when it still accounts for 43.9% of power generation. By contrast, coal power begins downscaling significantly on a large scale after 2035 and is almost completely phased out in 2050 under the CN2050 scenario. The changes in the capacity mix are shown in Supplementary Fig. 9. In the NDC and GM2.0 scenarios, the coal power capacity remains 910.0 and 155.9 GW, respectively, in 2050. Coal-power generation with CCS does not play an important role in the mix due to the high projected costs for $CO_2$ capture.

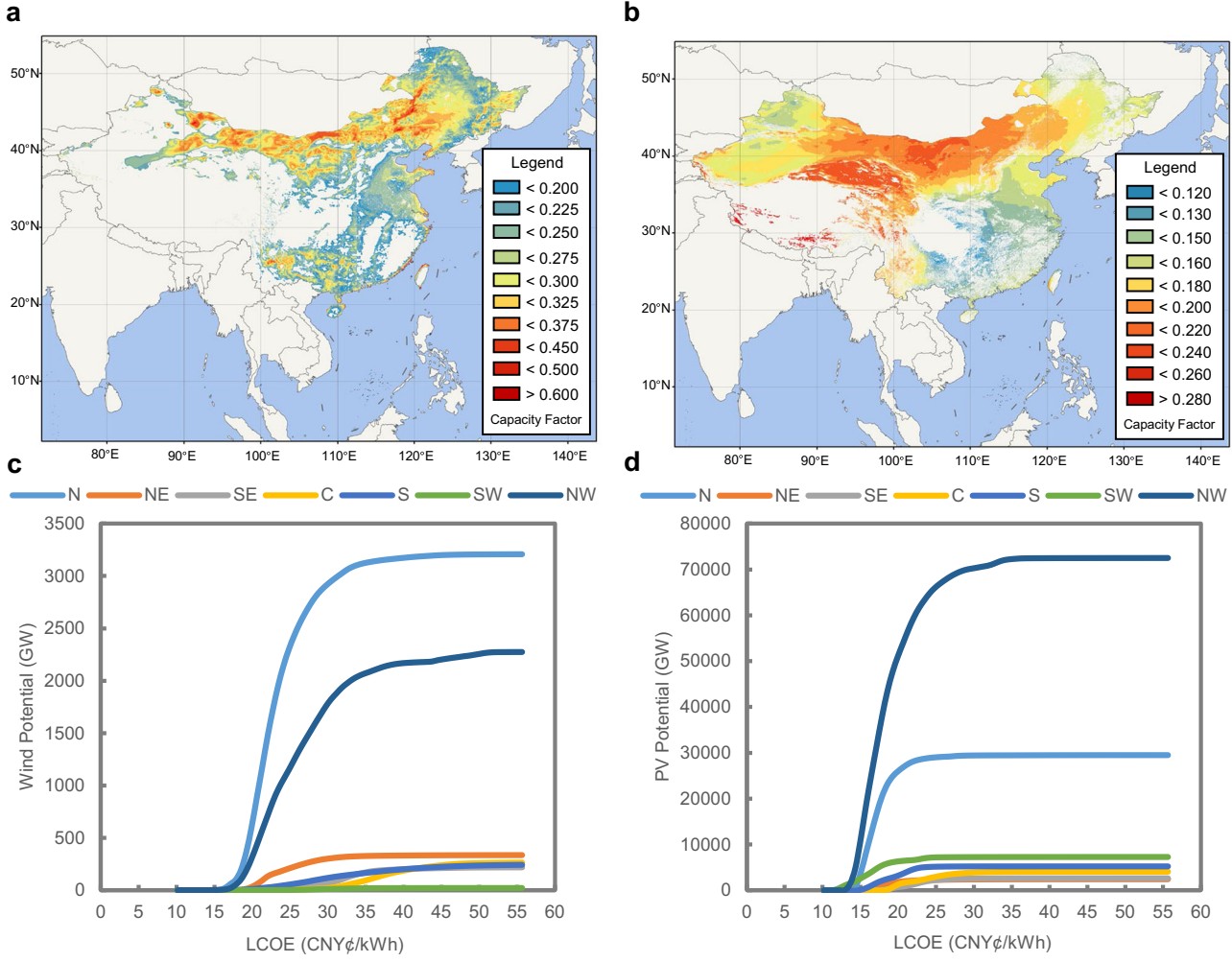

**Fig. 2 Spatial variable renewable energy (VRE) capacity factors across China and regional VRE supply curves. a** Wind capacity factors across China. **b** PV capacity factors across China. **c** Regional wind power supply curves. **d** Regional PV power supply curves. The wind and PV supply curves are presented for seven regions: North China (N), Northeast China (NE), East China (E), Central China (C), South China (S), Southwest China (SW), and Southeast China (SE). See Supplementary Figs. 7 and 8 for VRE supply curves for each province. Note that the levelized cost of energy (LCOE) and supply curves are calculated based on the forecast of capital cost per kW in 2035.

VRE units, specifically PV and wind, would be primarily deployed to offset the decline in coal power. For the BAU scenario, VRE would be developed only in later years, a result of its growing unsubsidized cost competitiveness, with a total capacity of 2485.9 GW in 2050, accounting for 32.8% of all power generation. For the CN2050 scenario, the VRE penetration reaches 72.8% in 2050 at 3579.0 GW of PV and 2247.1 GW of wind (ten times the VRE capacity in 2020). The installed capacities of VRE for the NDC and GM2.0 scenarios by 2050 are 4121.8 and 5247.7 GW, respectively. Figure 5 presents the provincial capacity mix in 2050 under the CN2050 scenario. Areas with large-scale onshore wind power capacities include in Gansu, Xinjiang, and Inner Mongolia, and a small amount of wind power is located in the northeastern and coastal provinces. Offshore wind power is concentrated in the three coastal load centres of Shandong, the Yangtze River Delta, and Guangdong. The disparities in installed PV capacities among provinces are not as significant as those for wind power because the average PV LCOEs are more similar across provinces. Although the VRE capital costs per kW are expected to continue to decrease in the future[8], the investments required to expand the total installed capacity will increase the electricity supply costs.

Power system security and stability require auxiliary services from flexible generation resources to enable the integration of VRE. Currently, such services are mainly provided by coal-power units, although this cannot continue to be the case under carbon neutrality. With the forced withdrawal of coal power, auxiliary services must be provided by natural gas units because of their relatively low rate of carbon emissions and generation flexibility. The NDC scenario has a relatively low gas capacity in 2050 at 137.4 GW, because considerable coal-power capacity remains. Under the GM2.0 scenario, the gas capacity increases from 111 to 449.4 GW by 2050. The installed capacity in CN2050 is slightly smaller, at 389.1 GW. To achieve carbon neutrality in CN2050, the emissions of these gas units must be offset by negative emissions achieved by generation using biomass energy with CCS (BECCS). Due to its highly variable operating cost (approximately twice that of coal power), biomass power is developed at a large scale only in the CN2050 scenario. Among the 187.5 GW of biomass power capacity, 43.4% is from BECCS units, contributing 182.7 million tonnes of negative carbon emissions per year. However, the supplies of natural gas and biomass fuel for power generation are relatively scarce in China, limiting the total corresponding capacities. These two types of units are mainly

**Table 1 Model scenario descriptions and settings.**

| | Description | Carbon emission budget during 2020–2050 (billion tonne) | Electricity load demands in 2050 (TWh) |
|---|---|---|---|
| Business-as-usual (BAU) | No carbon emission reduction targets are enforced. It is set for reference. | N/A | 12,300 |
| Nationally Determined Contribution (NDC) | Based on China's NDC goals and action plans under the Paris Agreement, the past low-carbon transition trends and policies in the power sector since the Paris Agreement was enacted are continued. | 104.4 | 12,300 |
| Global warming of 2.0 °C (GM2.0) | Oriented to achieve the IPCC goal of global average surface warming of no more than 2.0 °C, the carbon emissions from the power system are strictly limited while supplying the increasing electrified load of other industries. | 84.2 | 13,100 |
| Carbon Neutrality (CN2050) | Oriented to achieve carbon neutrality in the whole power system in 2050 while serving expanded electrified load from other industries. The CN2050 scenario corresponds to the requirements of the power system in China to meet the IPCC goal of global average surface warming of no more than 1.5 °C. | 66.2 | 14,860 |

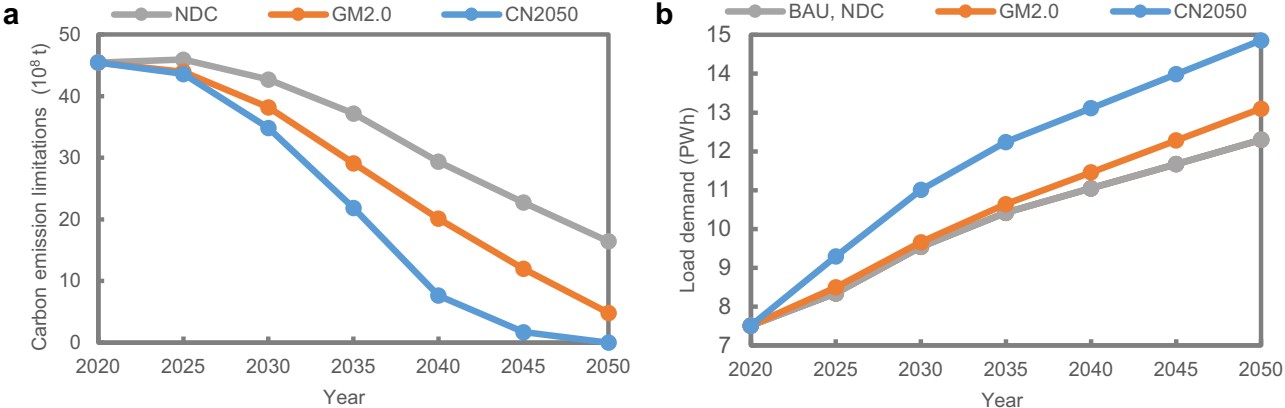

**Fig. 3 The annual carbon emissions and electricity consumption of the four different scenarios. a** Carbon emission limit trajectories over the 30-year planning period including Nationally Determined Contribution (NDC) scenario, Global Warming of 2.0 °C (GM2.0) scenario and Carbon neutrality (CN2050) scenario. Business-as-usual (BAU) scenario has no carbon emission limits. **b** Electricity load demand projections over the 30-year planning period. BAU and NDC scenarios have the same load demand trajectories.

distributed in coastal and central China because the primary fuel resources in those provinces are relatively abundant. The high fuel costs of natural gas and biomass in China would also contribute to increasing the electricity supply costs.

Energy storage systems (ESSs) play a critical role in accommodating high RE penetration and ensuring system security, as they are able to provide high-performance ancillary services including the contingency reserve, peak regulation, fast frequency regulation, and virtual inertia functions. Pumped hydro storage (PHS) and battery energy storage (BESS) are considered in the model. Due to the low VRE penetration and high battery capital costs in the early stages, BESSs are mainly deployed in the later stages of planning (after 2030). The total installed capacity of ESSs in GM2.0 and CN2050 is 967.7 and 1116.9 GW, respectively. The distribution of ESSs is roughly consistent with the distribution of VRE, primarily in areas with rich VRE resources and load centres. Concentrated solar power (CSP) plants with heat storage devices play a similar role as ESSs, while also providing RE generation resources. Because of limited solar irradiation conditions, CSPs are located mainly in northwest

China and Inner Mongolia, where particularly high-quality solar energy resources exist. The investment in ESSs and CSP reflects the external costs that are needed for VRE integration, which would be a considerable part of the electricity supply costs to achieve carbon neutrality.

High renewable penetration also increases the demand for long-distance transmission lines, and tighter interconnection is required to cope with the generation-load imbalance. Figure 6 presents the 2050 inter-regional and intra-regional power exchanges in the CN2050 scenario according to our results. In the CN2050 scenario, 378.3 GW of AC lines and 409.9 GW of DC lines would be newly constructed for interprovincial transmission. Interprovincial power delivery reaches 6052.73 TWh per year in 2050, which is 3.79 times the value in 2020 and 40.7% of the total annual demand. In comparison, the BAU scenario includes only 2777.9 TWh of interprovincial power delivery in 2050. The network structure of China's interprovincial transmission grid in the CN2050 scenario is presented in Supplementary Fig. 10. The distribution of power flow continues the current pattern of "from west to east" and "from north to south" in terms

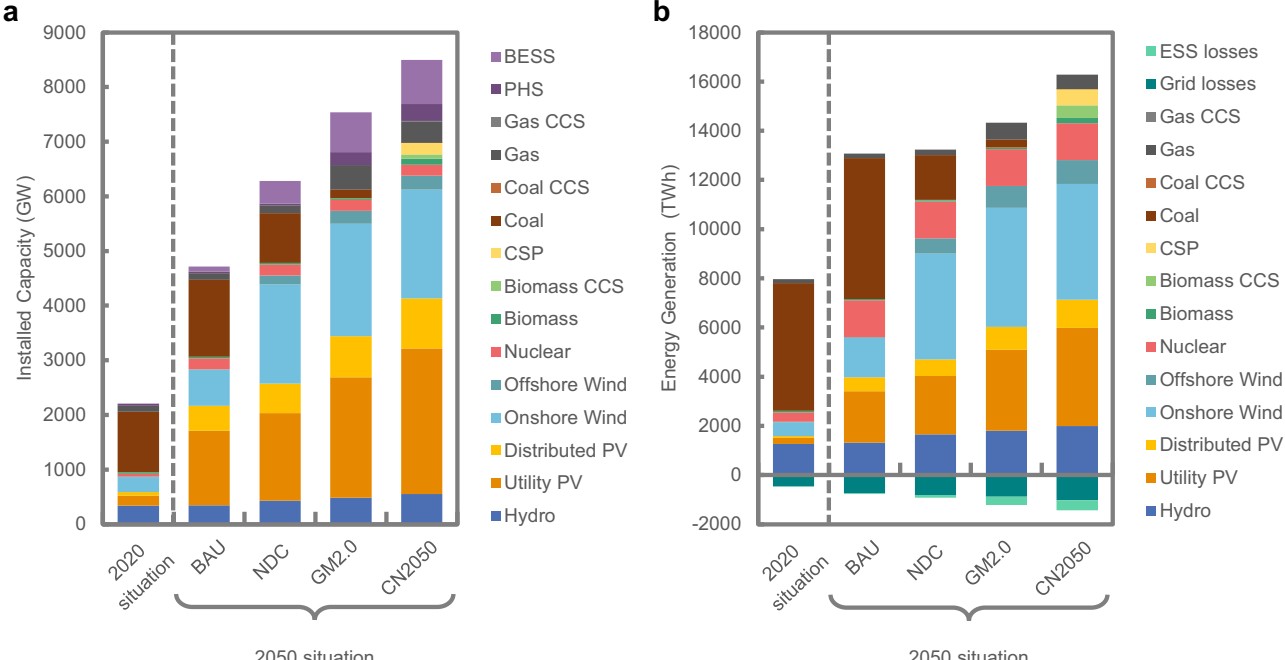

**Fig. 4 The generation mix in China under different carbon emission target scenarios. a** Comparison of the capacity mix between different carbon emission scenarios and the current status in 2020. Thirteen generation technologies are shown including hydro, utility photovoltaics (PV), distributed PV, onshore wind, offshore wind, nuclear, biomass power, concentrated solar power (CSP), coal power, gas power, and carbon capture and storage (CCS) units. CCS units include coal CCS, gas CCS, and biomass CCS. Two energy storage systems (ESSs) are shown including pumped hydro storage (PHS) and battery energy storage system (BESS). **b** Energy generated and lost by corresponding devices in 2050 under different scenarios and the current status in 2020.

of annual power exchanges. North and Northwest China are still the main power output regions, reflecting their abundant renewable energy potential. East and Central China are still the main receiving ends, with both receiving >800 TWh of external power annually. Although the annual power exchange pattern remains unchanged, the number of power flow reversals increase and the power exchanges among provinces mostly become bidirectional. Increases in power transmission will also increase the electricity supply costs.

**Costs of carbon neutrality.** Dramatic changes in power system morphology will inevitably cause changes in the electricity supply cost. We observe considerable but arguably affordable costs associated with transforming China's power system to support carbon neutrality. The total costs in the CN2050 and BAU scenarios over the 30-year period are 50.4 trillion CNY and 31.4 trillion CNY (present value in 2020, ~7.31 trillion USD and 4.55 trillion USD), respectively. The carbon neutrality goal results in an additional cost of 60.5% compared with that for BAU. The average annual additional costs are ~0.62% of China's GDP in 2020. It is a large but manageable value, especially recognizing that it does not account for the considerable external cobenefits of fossil fuel substitution, such as reduced health and agricultural damages from air pollution.

Figure 7 presents the electricity supply costs and marginal carbon prices for each scenario. In CN2050, the electricity supply cost increases by 9.6 CNY¢/kWh, or 19.9%, over the 30 years to 57.9 CNY¢/kWh (8.40 USD¢/kWh) in 2050. The process is not linear and can be roughly split into three stages. During the first five years, the cost curve is relatively flat, rising only 1.1 CNY¢/kWh. Between 2025 and 2045, the cost soars 17.0%, reaching 57.8 CNY¢/kWh (8.38 USD¢/kWh). With carbon neutrality almost achieved in 2045, the cost growth again slows due to

further reduction in per-kW RE capital costs and the complete withdrawal of coal power in the last five years. This trajectory of the electricity supply costs is largely driven by the second "3060" target, to achieve carbon neutrality by 2050 in the power sector (and by 2060 for the entire economy), while the first target, to reach peak carbon emissions by 2030, has relatively little effect.

By comparison, electricity supply costs would decrease 6.3% from 2020 to 2050 under BAU without carbon controls. Annual carbon emissions would reach 5.62 billion tonnes in 2035, and then begin to fall because the capital costs of RE would become sufficiently low, thus promoting competition with conventional units, even when considering their external costs. Specifically, declines in the per-kW VRE capital costs of 35% would incentivize the construction of many RE units even without carbon controls. The electricity supply costs in the other two scenarios exhibit slight decreases in the first five years and increases in the later years because of the more stringent emission requirements. The additional costs of emission reduction in 2050 for NDC and GW2.0 are 1.9 and 5.0 CNY¢/kWh, respectively, compared with emissions in the BAU case.

The average carbon mitigation costs are the additional costs paid per tonne of carbon emissions between the two scenarios. The average carbon mitigation costs grow rapidly when strengthening the emission limits across the four scenarios as shown in Fig. 7. The average additional cost for carbon mitigation is 699.6 CNY/t (101.4 USD/t) between GM2.0 and CN2050. This value is 16.2 times the difference between BAU and NDC (43.1 CNY/t). The marginal carbon prices reflect the cost of carbon emission reductions under various emission limits and represent the gradient of total costs for carbon emission reductions. With carbon neutrality requirements, the marginal price reaches 1444.2 CNY/t (209.40 USD/t) in 2050 under CN2050, and the 2020 actual carbon market-clearing price in China's carbon market is only 49 CNY/t (7.1 USD/t)[18]. As the emission reduction

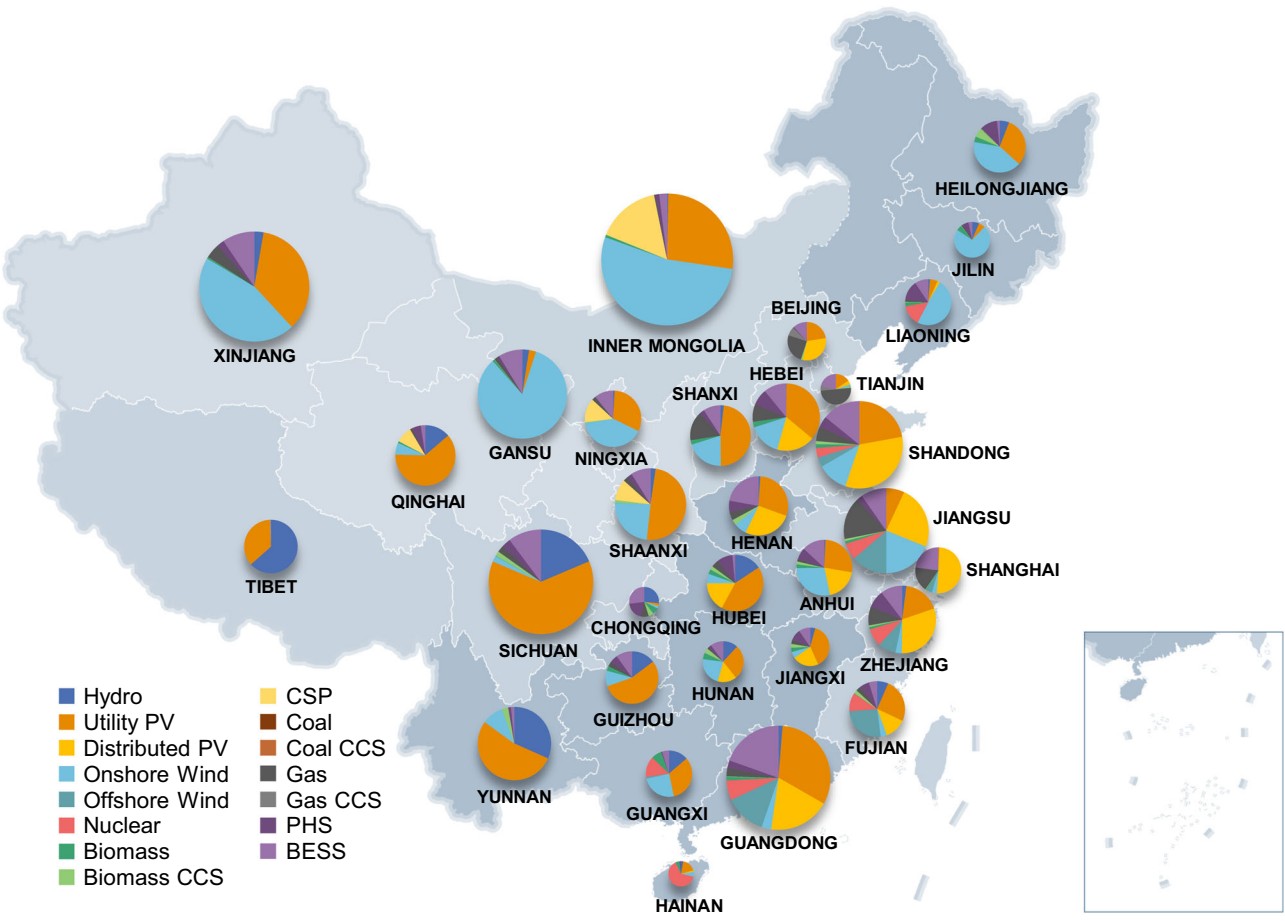

**Fig. 5 Provincial capacity mix in China in 2050 under carbon neutrality.** The size of the pie charts is proportional to the cumulative generation capacities for each province in 2050. Thirteen generation technologies are shown including hydro, utility photovoltaics (PV), distributed PV, onshore wind, offshore wind, nuclear, biomass power, concentrated solar power (CSP), coal power, gas power, and carbon capture and storage (CCS) units. CCS units include coal CCS, gas CCS, and biomass CCS. Two energy storage systems are shown including pumped hydro storage (PHS) and battery energy storage system (BESS).

requirements become more stringent, the same amount of emission reduction will result in a rapid increase in emission reduction costs. These results reflect the trade-off between low-carbon targets and the total costs. Hence, the less aggressive carbon reduction goals are much less expensive to achieve not only in aggregate terms but also on a per tonne basis, and the per-tonne expenditure increases rapidly under ambitious targets, such as those of CN2050.

**Composition changes of electricity supply costs.** The electricity supply cost structure shifts substantially over 30 years as changes are made to meet the carbon neutrality target. Figure 8 shows the changes in the composition of the electricity supply cost in the CN2050 scenario. The results for the other three scenarios are presented in Supplementary Fig. 11. The proportion of operating costs continues to decrease, and maintenance and capital costs continue to grow due to changes in the capacity mix reflecting the growth of RE and the retirement of thermal generation. The maintenance costs exceed the operating costs in 2050.

Figure 8 also shows the contributions of different factors to changes in the electricity supply cost. The changes are driven by three factors. The phasing out of coal power naturally causes a decrease in associated costs, and the installation of VRE results in an increase in capital and related external costs. A number of factors exert downward pressure on the electricity supply cost,

amounting in aggregate to 25.8 CNY¢/kWh. Almost all these factors are related to the reductions in coal-fired installed capacity and generation. Both the operating and capital costs for coal power decrease, with the latter accounting for the dominant share at 18.8 CNY¢/kWh.

Wind and PV power are the main driving factors in decarbonization. Since the two have zero operating costs, their capacity growth does not cause increases in operating costs. However, carbon neutrality requires an RE installed capacity of over 5.8 TW, reflecting in part the low capacity factors of wind and PV. Therefore, the capital and maintenance costs per kWh increase, even as the VRE capital cost per kW rapidly decreases. Under the assumptions of the CN2050 scenario, the supply cost increments contributed by VRE total 15.5 CNY¢/kWh, although the capital costs per kW decline by ~60%. The fact that VRE capital costs account for a considerable fraction of the supply costs illustrates the necessity of considering the supply curves for each province in detail. Absent consideration of the spatial distribution of LCOE, the costs in 2050 will be underestimated by 2.2 CNY¢/kWh. Explanations of the calculations and method used for comparison are presented in Supplementary Note 2.

Security and stability constraints in a power system with a very high RE capacity share require flexible generation resources and regional network expansion. Adding gas power, biomass power, ESSs, CSP and transmission expansion results in an additional supply cost increases of 18.4 CNY¢/kWh. This increase is a result

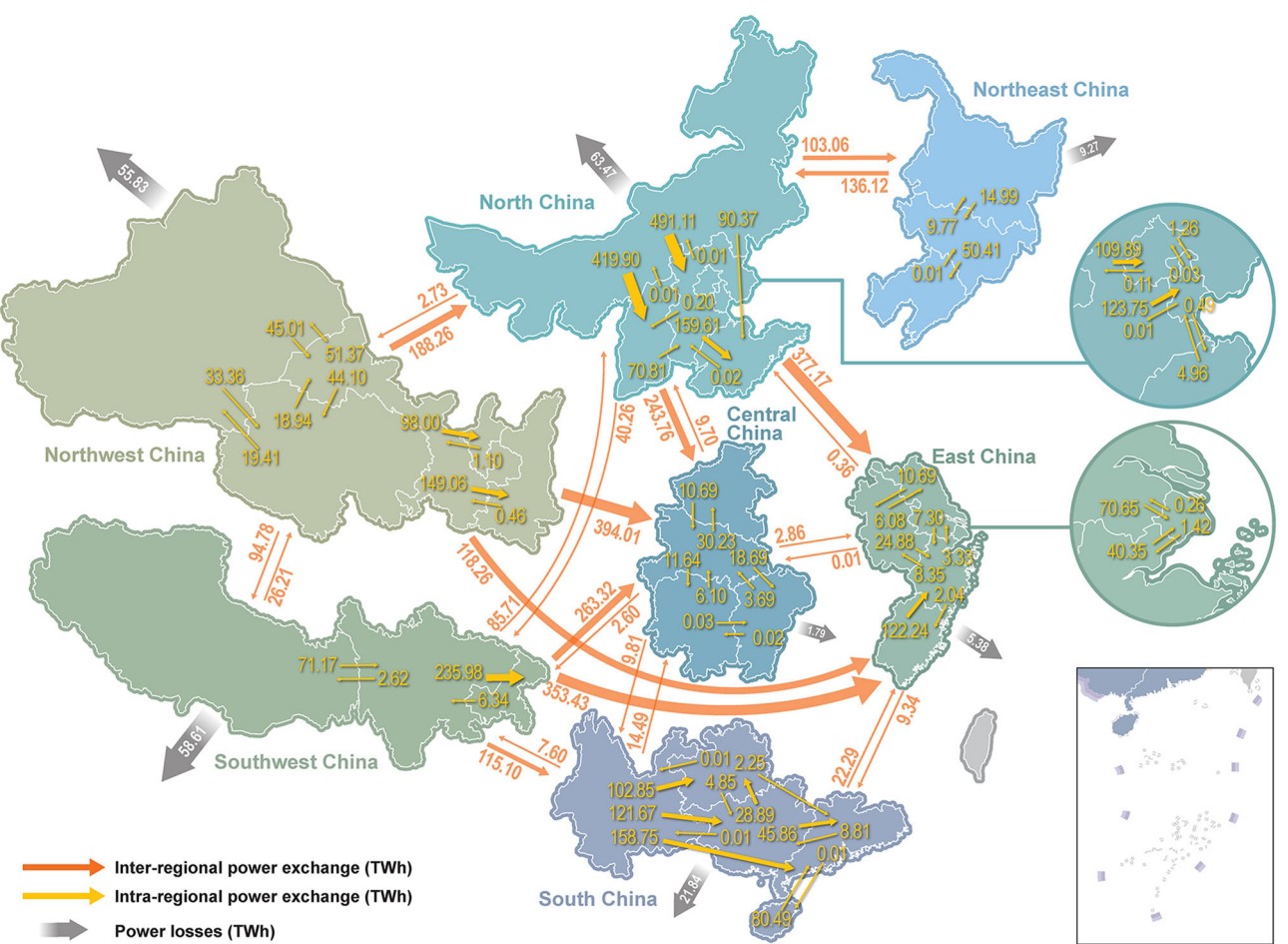

**Fig. 6 Power flow between and within seven regions in 2050 under carbon neutrality.** The numbers next to the orange and yellow arrows correspond to the annual inter-regional power exchange and intra-regional power exchange, respectively. The numbers in the grey arrows are the annual power losses of corresponding regions.

of the requirements for optimal development and operations considering the spatiotemporal imbalances and low-inertia characteristics of VRE.

**Electricity supply cost sensitivity**. A sensitivity analysis is conducted on the CN2050 scenario to consider the impact of the uncertainties of five key factors: RE and BESS capital costs, the load growth rate, security requirements, the RE and BESS manufacturing capability, and transmission capacity limits. We vary the parameters within the given intervals for each factor to assess their influence on the electricity supply cost increase in each stage (see Supplementary Note 4 for the sensitivity analysis settings and Fig. 9 for the results). Despite the influence of various uncertainties, increases in the electricity supply cost are almost inevitable as shown in Fig. 9. In most sensitivity analysis cases, the trajectory of cost changes is similar to that in the base case. In some cases, the costs slightly decline during the last period due to the maturity of generation technologies. These factors can be divided into two categories according to the periods in which they mainly affect the electricity supply costs. The RE and BESS capital costs, manufacturing capability, and security requirements most significantly impact the costs in the later stages of development, and the other factors impact the costs most significantly in the early to mid-term stages.

The uncertainties of RE and BESS capital costs (wind, PV, CSP, and BESS) result in the largest differences in the final electricity

supply cost at approximately ±6.5%. The electricity supply costs in 2050 under carbon neutrality vary from 54.3 to 61.9 CNY ¢/kWh as VRE capital cost increases from the low- to the high-penetration cases. Due to the accumulation of capital cost differences, such variation increases as the carbon emission limits tighten over the planning period. The results are expected considering the large contribution of RE capital costs to the total cost increases. However, the final supply cost would still rise by 6.0 CNY¢/kWh even in the case in which the cost decreases the fastest.

The uncertainties of security and inertia requirements, including power reserve constraints, spinning reserve constraints, and minimum inertia requirements, result in a ±3.0% difference in final supply costs. The variation increases as the penetration of VRE increases since VRE provides neither a robust reserve capacity nor inertia to the power system and reduces system security and stability. Stricter system security requirements would require increased investment in ESSs, CSP, and transmission lines to accommodate VRE, leading to rapidly rising costs. Without the consideration of three security requirements in the GTEP model, the electricity supply cost of achieving carbon neutrality in 2050 drops by 2.4 CNY¢/kWh.

The sensitivity of the transmission capacity limits exhibits a similar trend as that for the security requirements since transmission expansion is also an important means to ensure system security. Transmission capacity limits impact the electricity supply cost mainly after 2035, when the RE installed

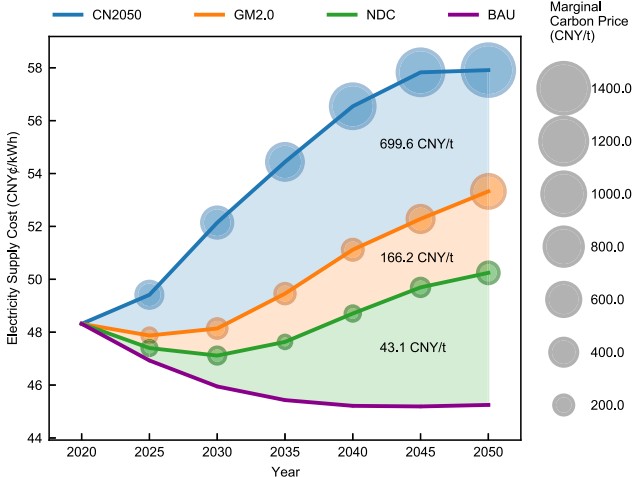

**Fig. 7 Electricity supply costs and carbon mitigation costs under four different scenarios.** The four scenarios include the Carbon neutrality (CN2050) scenario, Global Warming of 2.0 °C (GM2.0) scenario, Nationally Determined Contribution (NDC) scenario and Business-as-usual (BAU) scenario. The electricity supply cost is the average cost the power system must pay to supply per kWh of load demand. The marginal carbon price is the increased power system cost per tonne of carbon reduction based on the emission target each year. From the perspective of an optimization problem, the marginal carbon prices are the shadow prices of carbon emission constraints and can be obtained directly by solving the GTEP model. Their calculation does not need a reference case. The marginal carbon price in the BAU scenario is zero because no carbon emission constraints are considered. The number between each pair of curves denotes the average carbon mitigation cost per tonne between the more stringent scenario and the less stringent one. This value is numerically equal to the ratio of the difference in the total costs of the two scenarios and the difference between the carbon emission budget. The calculation methods are detailed in Supplementary Note 3.

capacity grows rapidly and greater electricity delivery is needed. The uncertainties of transmission capacity limits result in a ±1.7% difference in the electricity supply cost in 2050.

The impacts of the load growth rates and manufacturing capability exhibit different patterns, in which the influence on total costs is greater in the early to mid-term stages of system development. For load growth rates, the impact on cost decreases from a peak value of 4.1 CNY¢/kWh in 2035 to 1.3 CNY¢/kWh in 2050 because each additional kilowatt-hour of load demand in the initial stage would cause more carbon emissions and require more RE generation, which is relatively expensive at that time. Hence, the supply costs in the early stage are more sensitive to load demands than those in the completion stage.

The RE and BESS manufacturing capabilities represent the maximum capacities that can be newly installed each year, which are driven by the production capacities of RE generation and BESS units and related policies. Their sensitivity is highest in 2040 since the carbon emission limits are reduced by 78.1% from 2030 to 2040, and the ability to install RE generation and BESS becomes critical. Such capabilities become redundant in the late stage, so their sensitivity to electricity supply costs diminishes. However, when the manufacturing capacity is too low to produce enough VRE units more than a certain threshold over the future 30 years (~5000 GW according to our results), considerable investment in coal-based CCS units would need to meet the emission reduction goals, leading to gross increases in costs (see the outliers in Fig. 9 for the RE unit manufacturing capability).

Although the electricity supply costs will inevitably increase, there are some ways to reduce the increase according to the above

sensitivity analysis. The capital costs of RE and BESS units are the dominant factors affecting the electricity supply cost. Accelerating the cost decreasing by supporting the research and development of low-carbon generation and storage technologies would effectively reduce the transition costs associated with meeting the decarbonization target. In the early stage of the transition, it is critical to ensure a sufficient manufacturing capacity as much as possible. Moreover, reducing the total load demands by delaying the electrification process may reduce costs during this stage. However, this approach needs to be considered together with the low-carbon transition plans in other industries. In the later stage, a lack of interprovincial transmission capacity will lead to cost increases, and new transmission corridors will be needed. Reducing the reserve rates and minimum inertia limits also reduces the supply costs. However, this may not be allowed since ensuring system security is the first priority of power system planning and operation. The minimum value of the reserve rates and inertia limits to guarantee the system security stay unclear for future power systems, which requires further research on power system stability considering high renewable penetration.

## Discussion

Under the carbon neutrality goal, the morphology of China's power system will undergo tremendous changes. Our simulation results show that more than 5.8 TW of VRE units are needed to replace coal power. Large investments in ESSs, CSP units, and transmission lines will be required to meet the security requirements for the reserve capacity, minimum system inertia, and real-time power balance. In effect, an entirely new type of power system will be constructed, with wind and PV resources serving as the core (in terms of power generation) and various flexible generation resources serving as auxiliary service contributors. Moreover, the interprovincial grid interconnection must be gradually transformed from a simple power transmission channel to a platform supporting bidirectional energy sharing between regions with different generation resources.

Sufficient manufacturing capabilities for RE and ESS units are important for achieving a tremendous transformation in such a short time. From 2030 to 2045, the installed capacities of wind, PV, and ESS would need to increase annually by at least 91.2 GW, 139.4 GW, and 54.0 GW on average, respectively. For comparison, the average annual growth in the capacities of wind and PV in the past 5 years was 30.0 GW and 41.8 GW respectively. In addition, the total installed capacity of ESSs in China was just 35.6 GW at the end of 2020. Therefore, the current manufacturing of RE and ESS units is still insufficient for supporting the needed energy transformation, although the growth rate in China is the highest in the world. Ensuring the integrity of the supply chain and implementing appropriate policy incentives are critical steps for achieving the decarbonization of the power sector by 2050.

Rapid capacity mix changes cause considerable electricity supply cost increases. A complete set of power market mechanisms including spot, capacity, and ancillary markets, is urgently needed to reflect such changes in the electricity prices and fairly allocate the corresponding benefits. As mentioned, the electricity supply costs would increase by 9.6 CNY¢/kWh (1.39 USD¢/kWh) in an attempt to achieve carbon neutrality in 2050 even though as costs of RE technologies decrease following the current global trends. The cost increases will ultimately be paid by consumers through the renewable surcharges in electricity bills under China's current electricity pricing mechanism. The current administrative pricing method is rigid. For example, the RE surcharge of 1.9 CNY¢/kWh[19,20] has not been changed since 2018, despite a 49.1% increase in VRE installed capacities. Hence, a spot market

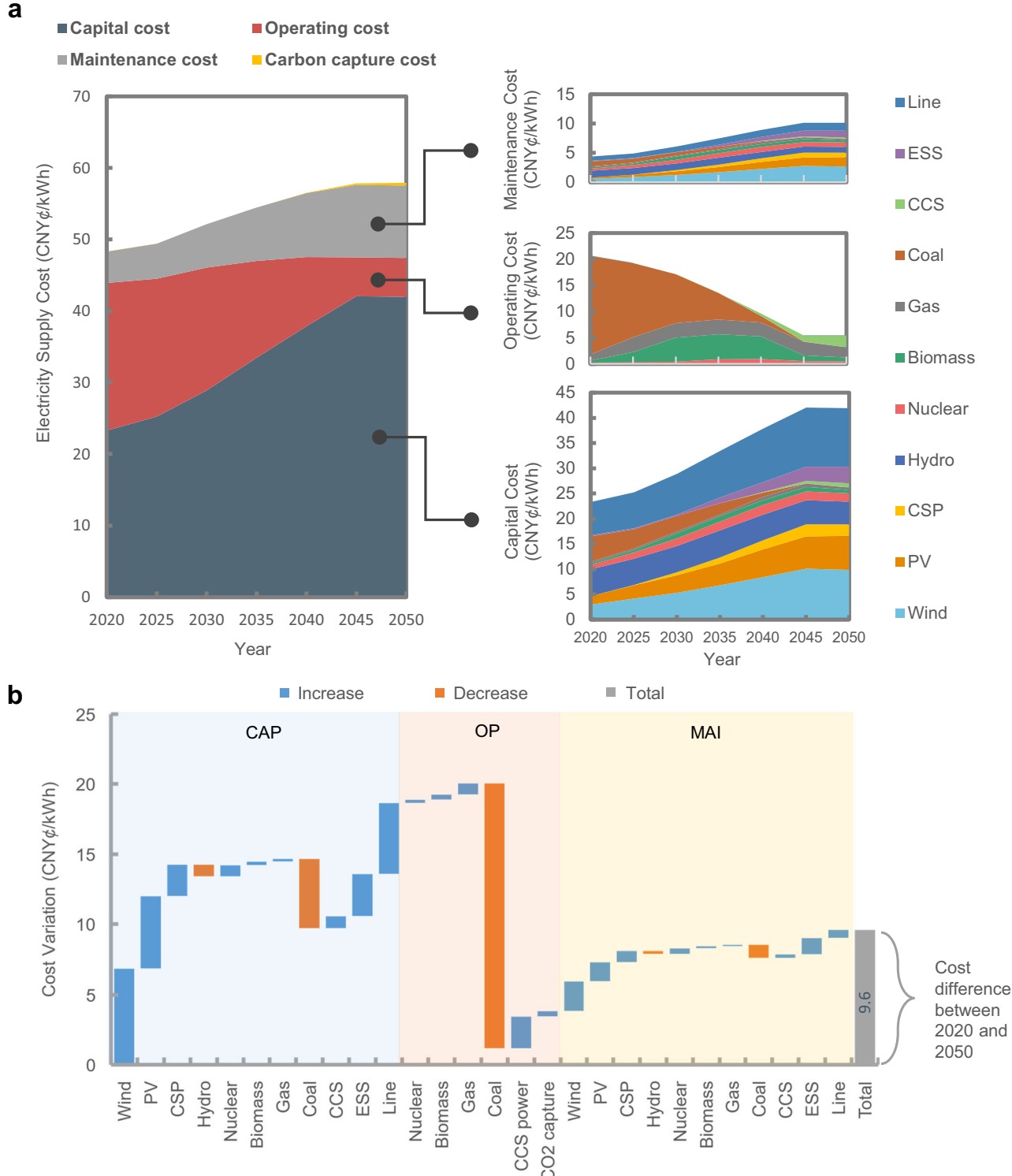

**Fig. 8 Composition variation of the electricity supply costs from 2020 to 2050 under the carbon neutrality goals. a** Electricity supply cost variation. The electricity supply cost consists of three parts: capital costs (CAP), operating costs (OP), and maintenance costs (MAI). Note that capital costs here are the one-time expenses incurred for the manufacture and construction of new electrical equipment, including for generation and transmission, which may also be named investment costs or overnight costs in the reference literature. The total capital costs are annualized over the plant lifetime to calculate the annual capital costs (see the detailed calculation method in Supplementary Note 3). Operating costs here indicate the variable costs including fuel costs and variable operations and maintenance (O & M) costs. Maintenance costs here indicate the fixed O & M costs, which are the costs incurred whether or not the power plant is generating electricity during operations and maintenance. The cost changes for different devices are broken down including wind units, photovoltaics (PV) units, concentrated solar power (CSP), hydro units, nuclear units, biomass units, gas units, coal units, carbon capture and storage (CCS) units, energy storage systems (ESSs) and transmission lines. **b** Contributions to the cost changes of different components in the power system. Each bar denotes the cost difference between 2020 and 2050 for the corresponding component. For example, the first bar denotes the contribution of the increase in the capital cost of wind power to the net change in the total electricity supply cost.

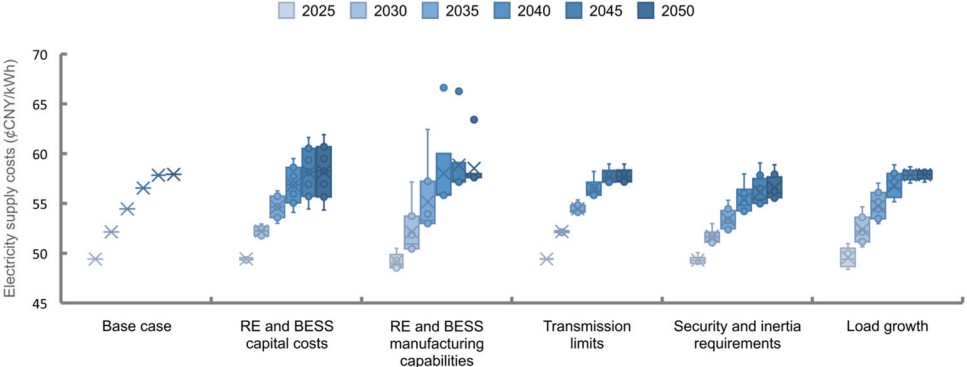

**Fig. 9 Sensitivity of the electricity supply costs to various assumptions.** Variations in the electricity supply costs are compared based on the costs in the Carbon Neutrality (CN2050) scenario considering different assumptions on renewable energy (RE) capital costs, RE unit manufacturing capabilities, transmission limits, reserve and inertia requirements, and load growth rates. Low and high scenarios are set for each factor. We vary the parameters within the given intervals with evenly sampled intermediate scenarios ($n = 7$). See Supplementary Note 4 for the detailed sensitivity analysis settings. The figure is a standard box plot. Each box has five horizontal lines representing the maximum value, the first quartile, the median, the third quartile and the minimum value from top to bottom. The cross points represent the average value, and the hollow circle points are the sampled value. Any point outside those lines is considered an outlier.

is required to obtain accurate and timely electricity prices. In addition, the electricity supply cost increases are derived from not only VRE capital costs but also from additional external costs to ensure system security due to the intermittency and low inertia of VRE. Flexible generation units, such as gas power, biomass power, CSP, and ESSs, are not major contributors to baseload power generation but support system security through their regulation ability and inertia. A capacity market and ancillary service market should be established to internalize the external costs of high RE penetration and to fairly allocate the benefits to these flexible units.

Our analysis indicates that achieving carbon neutrality in the power sector in China by 2050 is technically feasible but will increase the electricity supply costs significantly by 19.9%. Policy subsidies based on feed-in tariffs or renewable energy certificates should continuously be provided to newly emerging low-carbon generation and storage technologies, such as CSP and BESSs, in the early stages of their deployment. Historically, the Chinese government's feed-in tariffs for the wind and PV industries in the past 10 years have resulted in considerable cost reductions thus providing the foundation for the carbon neutrality transition. Similar policy supports remain vital for technological innovation and scaling for the newly emerging technologies to ensure the transition is not only successful but also cost-effective. Determining how policy formulation impacts the techno-economic performance and how to formulate concrete policies are topics that must be considered in future work.

## Methods

**GREAN platform.** To evaluate the factors that increase the electricity supply cost under high wind and solar penetration to achieve carbon neutrality in China's power system, we apply the integrated analysis platform Global Renewable-energy Exploitation ANalysis (GREAN) established by the Global Energy Interconnection Development and Cooperation Organization (GEIDCO). The key points for accurately assessing the LCOE are twofold: (1) detailed databases of weather, geographic and related infrastructure information are needed and (2) comprehensive assessment models that consider the costs of construction of flexible generation and grid interconnection under high wind and solar penetration are required.

The GREAN platform includes data on worldwide VRE resources, geographic characteristics, and human activities. The global VRE (wind and solar) energy resource data have a resolution of 9 km × 9 km. In addition, the wind energy resource data provided by Vortex[21] include wind speed, wind direction, air density, temperature, and other parameters. The solar energy resource data provided by SolarGIS[22] include global horizontal irradiance and direct normal irradiance. Combined with the characteristics of wind turbines and PV modules, these data are applied to evaluate provincial VRE capacity factors and generate the annual VRE

power output. The geographic data include a global land cover map, the distribution of major water bodies, terrain elevation data, and other information. Human behaviour-related data include the distribution of major conservation areas, information on the power grid and transportation infrastructure, and other information. Geographic information and human behaviour data are used to evaluate the availability of RE sites, the challenges of construction, and grid-connection costs. The specific data categories and sources of the database used in the GREAN platform are detailed in Supplementary Fig. 6 and Supplementary Tables 11–13.

Comprehensive assessment models are established based on the databases above to conduct systematic calculations of theoretical, technical, and economic VRE potentials. The theoretical VRE potential refers to the maximum developable capacity for VRE units considering only the VRE resource data. The technical VRE potential is obtained by excluding areas that are unsuitable for development due to poor resource endowment, disadvantageous geographic conditions, or policy prohibitions, such as conservation area designations. Based on the technical VRE potential, the economic VRE potential is calculated considering the construction constraints and grid connection costs and is then used to calculate the overall electricity supply cost under carbon control and neutrality targets. To evaluate the economic VRE potential, the GREAN platform first calculates the year-round hourly power output of wind and solar units in each 500 m × 500 m area based on the generation characteristics of 2.5 MW wind turbines with hub heights of 100 m, which meet IEC standards, and 300 W monocrystalline PV modules with fixed-tilt systems. The capacity factors are then calculated for each province. The LCOE is finally calculated considering the grid connection cost and device transportation cost according to the distance between the construction location and related infrastructure. The supply curves of the VRE generation potential around China can be obtained based on the LCOE at a fine spatial resolution to reveal the different characteristics of the RE resources in different regions.

With all the key factors considered, the GREAN platform provides LCOE and supply curve estimates that are as close as possible to the actual values based on the data available. The technical details of the platform are presented in the references[23].

**Generation and transmission expansion model.** A generation and transmission expansion planning (GTEP) model is formulated to simulate the long-term development roadmap of China's power system under carbon emission reduction targets in the next 30 years. The multi-period planning model is structured at the provincial level with embedded operation simulation at an hourly scale. Hourly dispatch schedules based on our model in 2020 are presented in Supplementary Fig. 12. Hourly dispatch schedules in 2050 under the CN2050 scenario are presented in Supplementary Fig. 13. Mathematically, GTEP modelling involves solving an optimization problem to minimize the total costs associated with electricity generation, transmission, and low-carbon transformation in given planning periods. Specifically, the total cost consists of capital, maintenance, and operating costs. Long-term investment schemes and short-term dispatch schedules are optimized simultaneously, and are subject to two sets of constraints: planning and operational constraints. The planning constraints represent resource limits and environmental policy regulations at the macro-level. Construction and retirement decisions for various generation units, ESSs, and transmission lines in different stages are made within the feasible range of the planning constraints. Because of the considerable impact of VRE capital costs on electricity supply costs, the dynamic changes in LCOE with increasing installed capacity are considered in the GTEP model. The VRE supply curves provided by the

GREAN platform are integrated into the optimization model in a piecewise manner for every province (see Supplementary Note 2). The operational constraints describe the daily power system dispatch at the micro-level with generation and transmission operational characteristics modelled from the bottom up. Eleven kinds of generation technologies are considered in the GTEP model: wind, PV, CSP, nuclear, hydro, biomass, gas, coal, Biomass-CCS (BECCS), Gas-CCS, and Coal-CCS. Two kinds of ESSs, pumped hydro and battery storage, are also considered. Both high-voltage direct current (HVDC) and high-voltage alternating current (HVAC) transmission technologies are considered as available candidate lines for expansion. The power system is operated under operating rules for power balancing and spinning reserve requirements, among other factors, while pursuing the minimum total operating cost. The mathematical expressions of these constraints, the code implementation method, and the solution algorithms are detailed in Supplementary Note 1. Further information on the costs of electricity generation technologies, including gas power, ESS, and CCS technologies is given in Supplementary Tables 14–16. The parameters of the transmission technologies are presented in Supplementary Tables 17–20.

The settings in the GTEP model for China's current power system are extracted from real-world data at the end of 2020 for each province, and generation mix, capacity factor, load demand, and interprovincial transmission capacity data are included[24]. The per-kW capital cost projections for generation technologies in the GTEP model refer to the results of NREL's annual technology baseline (ATB) model published in 2021[8], which includes three trajectories: conservative, moderate, and advanced. The per-kW capital costs in the NDC, GM2.0 and CN2050 scenarios follow moderate trajectories from 2020 to 2050. We conduct a sensitivity analysis of RE capital costs, where the NREL conservative projection is set as the high-cost case and the advanced projection represents the low-cost case (see Supplementary Fig. 2). Note that we did not directly use the US-specific $/kW costs provided by ATB reports in our model. We extracted the percentage trajectories from the ATB 2021 projections and applied them to our model. The $/kW (technically CNY/kW) values for the capital costs of generation and transmission units are based on reports from the China Electricity Council and China Electric Power Planning and Engineering Institute[2,5]. The specific values are shown in the fifth column of Supplementary Table 14. The capital costs and other technical parameters are set to the actual values for China's power system. We multiply the China-specific values by the percentage trajectories to obtain the CNY/kW cost trajectories over 30 years.

Load growth rates over the planning period are derived from the Research Report on China's "14th Five-Year" Power Development Plan[25]and the Report on China's Long-term Low-carbon Development Strategy and Pathway[3]. The load growth rate highly depends on the degree of electrification of other industrial sectors under different carbon emission reduction targets. The electrification of industrial loads will mainly come from industrial electric boilers, metallurgical electric furnaces, and auxiliary electric motors. The electrification of residential loads will mainly come from air conditioning loads, electric heating, and electric vehicles. There is an inherent error between the projected load demands and the actual situation. Hence, a sensitivity analysis is also conducted for load growth with 10% fluctuations (see Supplementary Table 1). Note that the specific load components such as electric vehicles and heat pumps are not precisely modelled in the GTEP model. The load demands are modelled in GTEP as boundary conditions that require a real-time balance between power generation and power consumption.

**Local grid development projection**. The capital costs and maintenance costs of the transmission and distribution system in each province account for a considerable portion of the total cost of electricity, reaching approximately 15% ~ 20% according to the previous research[6]. Such local grid planning is not directly optimized in the GTEP model because of the inaccessibility of data and calculation issues related to varying spatial resolutions. Hence, we project the investment of local network expansion in an ex-post fashion based on the historical local network capacity and the planning results of the GTEP model.

We collect annual installed transformer capacity and transmission line length data at each voltage level (750, 500, 330, and 220 kV) in each province from 2008 to 2018[26]. The historical annual electricity load demands and local power generation levels are obtained and used to regress the corresponding relationships with installed transformer capacity and transmission line length. To avoid the negative effects of abnormal historical data on the regression results in several small provinces, the provinces are grouped into seven regions, and the projection is conducted at a regional scale. A linear regression model with L1 and L2 regularizers is used as the regression model[27]. The regression method is implemented using the sklearn package in Python[28]. The fitted model is then used to estimate the local transmission investment based on the electricity load demands and power generation planning results in the next 30 years obtained from the GTEP model. The goodness of fit for the training set and the projection results are presented in Supplementary Tables 7–10. For distribution networks, investment below 110 kV is estimated based on the proportion of total capital costs in distribution network equipment with respect to the total capital costs of the entire power grid, which is based on data from the China Electric Council (CEC) yearly reports[2]. The electricity supply costs caused by local network investment and maintenance are 11.9 CNY¢/kWh (20.5%) in 2050 under the CN2050 scenario.

The capacity factors of the entire capacity mix will decrease as VRE penetration increases, which will further reduce the utilization rate of network equipment[29]. To consider such impacts of changes in the capacity mix on local power grids, we modify the local grid investment based on historical data according to the changes in the capacity factors of the local capacity mix. Assuming that the utilization rate of power grid equipment decreases by the same rate as the power supply capacity factor, the modification formula is as follows:

$$\tilde{C}_{r,y}^{\text{local}} = \frac{\text{cf}_{r,0}}{\text{cf}_{r,y}} C_{r,y}^{\text{local}}, \forall r, y \tag{1}$$

where $C_{r,y}^{\text{local}}$ and $\tilde{C}_{r,y}^{\text{local}}$ denote the projected and modified local network investments in region $r$ in planning stage $y$, respectively. $\text{cf}_{r,0}$ and $\text{cf}_{r,y}$ denote the current capacity factors and capacity factors in planning stage $y$ for the capacity mix in region $r$, respectively. The capital costs and the maintenance costs for the local grid network are calculated based on the modified network capacities.

**Reporting summary**. Further information on research design is available in the Nature Research Reporting Summary linked to this article.

## Data availability

The per-kW capital cost projections for generation technologies used in this study are available from the National Renewable Energy Laboratory [https://atb.nrel.gov/electricity/2021/data]. The wind energy resource data are provided by Vortex [https://vortexfdc.com/]. The solar energy resource data are provided by SolarGIS [https://solargis.com/docs]. The source data underlying all the figures in the main article and supplementary information are provided as a Source Data file and deposited into a public data repository[30]. All data used for this study are available in Supplementary Information, the Source Data file, the public data repository[30] and cited from publicly available sources.

## Code availability

Codes used in MATLAB and Python for this study are available from the authors upon reasonable request.

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

## Acknowledgements

This work was supported by the National Natural Science Foundation of China (No. 52130702, No. 52177093 and No. 51907100 to C.K., N.Z., E.D., and Z.Z.) and the National Natural Science Foundation of China (No. 72025401 to X.L.)

## Author contributions

Z.Z., E.D., N.Z., and C.K. conceived and designed the research. Z.Z., E.D., and N.Z. formulated the theoretical model. Z.Z. conducted the code implements and carried out the simulations. Z.Z., N.Z., C.P.N., and X.L. conducted the data analysis and led the writing of the paper. N.Z., C.P.N., and X.L. wrote the introduction. Z.Z., E.D., J.X., and J.W. contributed to the data collection and figure drawing. N.Z. and C.K. conducted the policy analysis. Z.Z., E.D., N.Z., C.P.N., X.L., J.X., J.W., and C.K. contributed to the discussions on the framework and the editing of this article.

## Competing interests

The authors declare no competing interests.
