## [Peer Review File · Nature Communications]

Reviewer comments, first round review –

Reviewer #1 (Remarks to the Author):

1. There are quite a few typos in the manuscript. Need to be corrected. Also, the paper needs an English language copy edit. There are several cases of awkward English use.
2. Do authors use DC OPF problem formulation? If so, how are security and inertia constraints factored into a DC OPF model? What is the transmission detail? How are these constraints applied in absence of a full transmission network?
3. There is a typo in table 1. Should be '000 TWh.
4. Authors say they use ATB 2020 projections for capital costs. But ATB projects US-specific costs. Using the same \$/kW number in China does not make sense. Authors provide no justification of using the US costs in China or how they have been corrected for use in China. Second, according to ATB, the average LCOE in 2035 of a coal power plant is \$70-100/MWh while that of solar and wind power is \$15-20/MWh. The coal power plant variable costs in US and China are comparable (in fact, US costs are somewhat lower). Therefore, I am surprised to see how authors are showing over 400 GW of new coal in the BAU case. Are coal capacity additions driven mainly by the system security constraints or deployment constraints on RE?
5. Apparently, from the SI, it appears that the coal capital costs have been taken to be 4106 CNY/kW, which is equivalent to ~\$650/kW. Using US-based costs for clean technologies (without adjusting for better capacity factors) and Chinese costs for coal based power is not right.
6. For wind, I believe the study is using US capital costs (which are for much higher hub heights and rotor diameters) but still considering lower hub heights and rotor diameters in China - which explains the low capacity factors. For example, ATB reported capacity factors are in excess of 45-50%, nearly double what the authors use in their study. Resource quality alone cannot explain the differences in capacity factors in the US and China because wind speeds at >100m hub heights in certain regions in China are comparable to some of the best wind sites in the US.
7. Similarly for solar, authors use a capacity factor of ~15-18%, but ATB reported capacity factors are in excess of 25-30% due to inverter overloading etc - which would also explain somewhat higher capital cost in ATB.
8. Also, NREL has published ATB 2021, which shows even lower renewable prices.
9. Points 4-7 make me question the study's incremental cost numbers and other results.
10. What are the total emissions from the power sector by 2050 and key intermediate years?
11. The discussion section only summarizes the key results that are already presented. There are no concrete conclusions drawn per se and no discussion on the policy directions needed.

Reviewer #2 (Remarks to the Author):

This study comprehensively evaluate the cost increments to achieve the carbon neutrality in China electricity sector. This study use a bottom-up energy system model specialized for the capacity and transmission planning in the electricity sector. Even though the authors use lots of data and complex modelling framework, the manuscript is well-organized and well-written to follow easily.

Most of analysis results make sense and several implications are expectable. However, I agree that the main contribution of this kind of study is to deliver the implications with quantitative and specific numbers.

Nevertheless, I think that authors can improve the manuscript based on my comments:

1. The suggested model (GTEP model) seems to be a conventional power system planning model. If exist, please emphasize the new features of the model compared to previous studies.
2. Even though this study focus on the costs in the electricity sector, the main manuscript and the supplementary notes do not provide the full information about the costs of electricity generation technologies. Please provide further information on this, including gas and CCS technologies.
3. Why the electricity demand are different among scenarios in Table 1? How to derive these?
4. Please provide more comprehensive discussions about the sensitivity analysis

One minor comment:

Based on my experience, Figure 3(a) is called "capacity mix" and Figure 3(b) is called "generation mix" generally.

Reviewer #3 (Remarks to the Author):

Results are noteworthy and I think the work will be of significance to the field. The paper is similar He Gang et al "Rapid cost decrease..." among others. Consider providing a comparison with such papers, on both methods and results.

Your work supports the claims and conclusions you present but some assumptions like carbon reduction trajectory might be better if expressed explicitly in the body of the paper.

Data analysis and discussion at times seem a bit disconnected, positioning discussion and tables/figures closer would be beneficial. In addition, it seems that there is a bit of lack of clarity on capital/investment/overnight costs. Either explicitly say that they are interchangeable or define them and stick to one of the terms.

The methodology is sound and other papers use similar planning and optimization models.

One detail I do not see being provided in the supplemental information is load estimates at the provincial levels for each of the scenarios.

Response to Reviewers' Comments on “Cost Increase in the Electricity Supply to Achieve Carbon Neutrality in China”

November 26, 2021

First of all, we would like to thank the reviewers for the time and effort that you have put into reviewing the previous version of the manuscript. Thank you all very much for your careful review and valuable comments. The suggestions and comments have enabled us to improve the quality of our paper.

Our responses to the comments and questions are given directly below them in this letter. We have tried our best to improve the presentation of the paper. All the modifications in the article have been **highlighted in grey**. The major improvements are summarized as follows:

- The projection trajectories of low-carbon generation and storage technologies have been updated to Annual Technology Baseline (ATB) 2021[1]. Other parameters such as VRE capital costs have also been updated to the current status of China in 2020 based on the recently published reports [2, 3, 4]. We believe the updated input data are the most timely data we could access currently. We solved the planning problem again and recalculated the electricity supply cost based on the updated input data as suggested by reviewers.
- Assumptions on transmission line modelling method and generation technology parameters in China are justified based on existing related literature and authorized technical reports.
- The detailed data on the costs of generation technologies, the capacity factor of VRE units, and provincial load demands were added to the Supplementary Information (SI). The data sources for some key parameters are clarified in the paper.
- The source data of all the figures in the main article were uploaded to a persistent repository for researchers[5]. The current link presented in the response letter is now shared privately and its DOI is 10.6084/m9.figshare.16929340. The DOI will become active when the item is officially published.
- More comprehensive discussions about the sensitivity analysis and policy implications are supplemented in the main article.
- The new features of our proposed model compared with existing literature are emphasized in the main article. Meanwhile, the advantages of our model compared with existing methods are summarized in Supplementary Note 6.
- The revised manuscript was polished by “Nature Research Editing Service” and the native English speaker in our author group to avoid awkward English use. We proofread the manuscript again and corrected the typos based on the reviewers' suggestions.

Please see the revised manuscripts and the following responses for details.

Response to the Reviewers

Reviewer #1

Comment 1: There are quite a few typos in the manuscript. Need to be corrected. Also, the paper needs an English language copy edit. There are several cases of awkward English use.

Reply: Thank you for your detailed review. We corrected the typos found out by reviewers and proofread the manuscript again. The revised manuscript was polished by “Nature Research Editing Service”, a professional English language editing service from the publisher of Nature. The editing certificate is attached at the end of the response letter. The native English speaker in our author group also edited the whole manuscript to make the word choices and phrase constructions as exact as possible. Please see the revised manuscript for details.

Comment 2: Do authors use DC OPF problem formulation? If so, how are security and inertia constraints factored into a DC OPF model? What is the transmission detail? How are these constraints applied in absence of a full transmission network?

Reply: Thank you for your insightful comment. We use the pipeline model (also known as the “transportation model”) rather than the DC OPF formulation to model the transmission network between provinces. The modelling method derives from Ref. [6], where the power flow on the transmission line can be deployed freely within the rated capacity. The details on the transmission model are presented in Eq. (27) - Eq. (32) of Supplementary Information(SI). Such a formulation is reasonable for our GTEP model. In our model, each bus corresponds to an aggregation of a provincial power grid, not a real bus in the power system as stated in Supplementary Note 1.1. Thus, the free control of power flow on the AC transmission lines can be achieved by the line switch operation or the coordinated dispatch of reactive and active power within the province grid. For DC transmission lines, free control is their inherent advantage thanks to the power-electronic control technologies. Hence, the power exchange across provinces may not follow the DC power flow model. The pipeline model is more reasonable than DC OPF when considering the national transmission power system planning. DC OPF formulation is more suitable for the planning of a single provincial transmission grid or a city-level transmission grid.

Conducting the pipeline model avoids the introduction of binary variables, which considerably reduces the complexity of the GTEP model. The modelling of power losses is also smoother when applying the pipeline model. Currently, many planning studies for national or regional power systems have adopted the pipeline model[7, 8, 9, 10, 11, 12, 13]. We emphasized the transmission modelling method and the reasons in both the main article and the SI. Please see the “Transmission Line Model” subsection in Supplementary Note 1.4 for details.

We had considered the roles of the transmission network in the security constraints. It is a writing mistake that we did not present how it is done clearly in the previous version of SI. We have corrected the related mathematical formulation. The modified constraints of “Power Reserve Requirements” is as follows:

$$\sum_{m \in \{\Omega_r^{\text{GP}}, \Omega_r^{\text{ESS}}\}} r_m U_{m,r,y} + \sum_{m \in \Omega_r^{\text{DC,to}}} U_{m,y}^{\text{DC}} \geq (1 + r s_r) \left(\max_{d,t} L_{r,y,d,t} + \sum_{m \in \Omega_r^{\text{DC,from}}} U_{m,y}^{\text{DC}} \right), \forall r, y \quad (1)$$

where the item on the left of the inequality sign represents the power reserve that can be provided by the local generators and the transmission grid, and the item on the right represents the local reserve demand in the region r (Note that a region corresponds to a province here). $r_m U_{m,r,y}$ denotes the power reserve capacities provided by the local generation plants of technology m in region r at stage y . $U_{m,y}^{\text{DC}}$ denotes the the power reserve capacities by the DC transmission lines. $\Omega_r^{\text{DC,to}}$ denotes the subset of the whole DC transmission lines whose receiving end is region r . The local reserve demand is proportional to the peak load $\max_{d,t} L_{r,y,d,t}$ and the reserve of other regions brought by the DC transmission line. $\Omega_r^{\text{DC,from}}$ denotes the subset of the whole DC transmission lines whose sending end is region r . $r s_r$ is the required reserve rate in region r . In practical engineering, the sending end deploys dedicated power plants for the DC lines and the receiving

end regards the DC lines as dispatchable units. Hence, DC transmission lines can transfer the power reserve capacities between the provinces in our model. AC transmission lines cannot transfer the reserve because they cannot be dispatched as flexibly as the DC lines. Hence, AC transmission lines are not modelled in the constraints of “Power Reserve Requirements”

The modified constraints of “Spinning Reserve Requirements during Operation” is expressed in Eq.(2)-(3). The modification is similar to “Power Reserve Requirements”.

$$\sum_{m \in \Omega^{GP}, \Omega^{ESS}} P_{m,r,y,d,t}^{hot} + \sum_{m \in \Omega_r^{DC,to}} P_{m,r,y,d,t}^{hot} \geq hr_r^{Load} \cdot \left(L_{r,y,d,t} + \sum_{m \in \Omega_r^{DC,from}} P_{m,r,y,d,t}^{hot} \right) + hr_r^{Wind} \cdot \sum_{m \in \Omega^{Wind}} P_{m,r,y,d,t}^{Wind} + hr_r^{PV} \cdot \sum_{m \in \Omega^{PV}} P_{m,r,y,d,t}^{PV}, \forall r, d, y, t \quad (2)$$

$$0 \leq P_{m,r,y,d,t}^{hot} \leq U_{m,y}^{DC} - f_{m,y,d,t}^{DC,to}, \forall m \in \Omega_r^{DC,to}, r, t, d, y \quad (3)$$

where $P_{m,r,y,d,t}^{hot}$ denotes the spinning reserve provided to the receiving end by DC transmission lines. The upper boundary of $P_{m,r,y,d,t}^{hot}$ is equal to the difference between the rated capacity and the current power flow. The spinning reserve demands are also transferred to the sending end as in Eq.(2). In our model, the spinning capacities indicate the capacities that flexibility units are able to be provided within 10 minutes. The dispatch of the AC line power flow is difficult to achieve within this time scale because it relies on the adjustment of the generator output and switching operation. Hence, AC lines are also not modelled in the constraints of “Spinning Reserve Requirements”.

The inertia of the power system mainly comes from the rotating parts of the local generators or the virtual inertia provided by the energy storage systems (ESSs). The transmission network cannot provide inertia itself. The long-distance transmission effect of the system inertia is still unclear and belongs to the field of power system transient analysis, which is beyond the scope of this paper. Therefore, we assume that the transmission network cannot transfer inertia between provinces on the transient time scale. We consider the inertia constraints for each province individually for system transient stability.

The transmission grids within each province are not integrated into our GTEP model explicitly because of the inaccessibility of data and calculation difficulties due to their spatial resolution. However, we project the investment of local network expansion based on the historical local network capacity and the planning results of the GTEP model. Moreover, the projected investment decisions are modified according to the changes in the capacity factors of the local generation mix during the decarbonization transition. The projection and the modification reasonably present the security requirements of the provincial transmission networks. Please see the “Methods” section for details.

Comment 3: *There is a typo in table 1. Should be '000 TWh.*

Reply: Thank you for your careful review. We have corrected the mistake. Please see Table 1 in the revised paper for details.

Comment 4: *Authors say they use ATB 2020 projections for capital costs. But ATB projects US-specific costs. Using the same \$/kW number in China does not make sense. Authors provide no justification of using the US costs in China or how they have been corrected for use in China. Second, according to ATB, the average LCOE in 2035 of a coal power plant is \$70-100/MWh while that of solar and wind power is \$15-20/MWh. The coal power plant variable costs in US and China are comparable (in fact, US costs are somewhat lower). Therefore, I am surprised to see how authors are showing over 400 GW of new coal in the BAU case. Are coal capacity additions driven mainly by the system security constraints or deployment constraints on RE?*

Reply: Thank you for your careful review. We did not use the US-specific \$/kW cost number in our model. We did not make this clear in the original text and caused a misunderstanding. It is the percentage of cost

Table 1: Comparison of LCOE between coal power and VRE in China

Unit type	Investment costs (CNY/kW)	Variable costs (CNY/kWh)	Maintenance costs (CNY/kW)	Lifetime (year)	Capacity factor (%)	LCOE reduction in 2035 (%)	Current LCOE (CNY/MWh)	LCOE in 2035 (CNY/MWh)
PV	4599	0	66.5	25	14.74%	45.14%	386.23 (\$55.9/MWh)	211.34 (\$30.6/MWh)
Wind	7600	0	146	25	23.78%	33.79%	411.89 (\$59.7/MWh)	272.70 (\$39.5/MWh)
Coal	4046	0.274	62	40	49.17%	5.98%	367.17 (\$53.2/MWh)	345.21 (\$50.1/MWh)

change, or more precisely its percentage trajectories, that we extracted from the ATB 2020 [14] projections and applied in our model. The \$/kW (technically CNY/kW) values for the investment costs of generation and transmission units are based on the authorized reports from China Electricity Council and China Electric Power Planning and Engineering Institute [15, 16]. The values are shown in the fifth column of Supplementary Table 14. They are China-specific and lower than that in ATB projections. We multiply the current China-specific values by the percentage trajectories to obtain the CNY/kW cost trajectories over 30 years.

Note that we have updated the percentage trajectories to ATB 2021 in the revised manuscript according to Comment 8 of Reviewer #1. Meanwhile, other parameters such as VRE capital costs have also been updated to the current status of China in 2020 based on the recently published reports [2, 3, 4]. The changes in the input data have minor impacts on the quantitative results of this article. Please see the details in the reply to Comment 8 and the revised manuscript.

We use the percentage trajectories provided by the ATB because there is no authorized long-term cost prediction of low-carbon generation and storage technologies in China. So we use the trend in the United State instead. The trend in the United States can represent the global trends. The changing trend of RE technology cost is supposed to be similar between the US and China since a great part of wind and PV devices deployed in the US are imported from China or China’s neighbouring countries[17, 18, 19]. The calculation method has been conducted in the existing literature on power system development beyond the US[9, 12, 20, 21]. The percentage trajectories of conventional units, i.e. thermal power and hydropower, refer to Ref.[22] which conducts the extrapolation method based on cost changes over the past 10 years. Due to the maturity, the capital costs of these generation units will not change significantly. The capital cost of hydropower will even rise slightly due to the increasing difficulty of construction. The data source and how it is applied are emphasized in the Method section of the revised paper to avoid potential misunderstanding. We understand that there is an inherent error between the projection data and the actual situation. That is why we conducted a sensitivity analysis on the capital costs of RE and ESSs.

The coal-power capacity increases from 1104.5 GW to 1409.4 GW in 2050 under the BAU scenario of the revised manuscript. In the previous manuscript, the coal-power capacity increases from 1104.5 GW to 1446.2 GW in 2050. The growth of coal-power capacity in the two versions is similar. To explain why considerable coal-power capacity is invested under the BAU scenario, we calculated the LCOE of coal power and variable renewable energy (VRE, indicating wind and PV) based on the China-specific data in 2020. The calculation method refers to Ref. [23]. The involved data and results are presented in Table 1, where the LCOE reduction in 2035 derives from the ATB 2021[1]. The capacity factors are the average value for all existing units in China [15]. The discount rate is set to 8%. As shown in Table 1, the current LCOE of VRE is close to coal power but still slightly higher than that of coal power due to their low capacity factors and short lifetime. In addition, the uncertainty and variability of VRE cause extra costs of peak regulation and reserve deployment for the whole power system. Thus, coal power is still the first choice of generation expansion during the early stage of planning as presented in Fig. 1(a). After 2035, the LCOE of VRE is lower than that of coal power and the installed capabilities of VRE start growing. Meanwhile, the coal-power capacities still increase because of the system security constraints including reserve constraints and inertial requirements.

Figure 1: Installed capacity and power generation of coal power and VRE during the planning period in the BAU scenarios

Without carbon emission limits in the BAU scenario, coal power is preferred to provide reserve and inertia compared with gas power and ESSs because of its low costs. The total energy generated by coal power decreases after 2035 though the capacities keep growing as shown in Fig. 1(b). This fact confirms that the role of coal power has shifted from power generation to system security supports gradually. In summary, the low capacity factor, externality costs of VRE, and the system security constraints together drive the new investment of coal power in the BAU scenario. Therefore, the investment of new coal power in the BAU scenario is sensible. We set no deployment constraints on RE in the BAU scenario.

In addition, it is worth mentioning that the average LCOE of coal power in ATB reports is higher (\$70-100/MW) because it involves the new type coal power plants such as Coal-IGCC and Coal-CCS. Both their investment costs and variable costs are much higher than the conventional coal power.

Comment 5: Apparently, from the SI, it appears that the coal capital costs have been taken to be 4106 CNY/kW, which is equivalent to ~\$650/kW. Using US-based costs for clean technologies (without adjusting for better capacity factors) and Chinese costs for coal based power is not right.

Reply: Thank you for your careful review. We did not use the US-based costs for clean technologies. All the parameters of generation and transmission technologies applied in our model are consistently China-based including investment costs and capacity factors. The current average investment costs of wind and PV power in the US are \$1400/kW and \$1300/kW, respectively [1]. The current average investment costs of wind and PV power in China are about 7600 CNY/kW (\$1101.9/kW) and 4599 CNY/kW (\$666.8/kW), respectively, as shown in Table 1 and Supplementary Table 14[2, 3]. The investment costs of VRE in China are lower.

We are sorry that the data source is not clarified clearly in the previous version and caused a misunderstanding. We emphasized the data source in the “Method” section of the revised paper to avoid potential misunderstanding. Please see the revised paper for details.

Comment 6: For wind, I believe the study is using US capital costs (which are for much higher hub heights and rotor diameters) but still considering lower hub heights and rotor diameters in China - which explains the low capacity factors. For example, ATB reported capacity factors are in excess of 45-50%, nearly double what the authors use in their study. Resource quality alone cannot explain the differences in capacity factors in the US and China because wind speeds at >100m hub heights in certain regions in China are comparable to some of the best wind sites in the US.

Reply: Thank you for your careful review. We did not use the US capital costs for wind power as mentioned in previous responses. The current average investment costs of wind power in the US are \$1400/kW according to ATB 2021[1]. The current average investment costs of wind power in China are about 7600 CNY/kW (\$1101.9/kW)[2, 3]. Both the investment costs and capacity factors used in the model are China-based. The capacity factors are based on the generation characteristics of 2.5 MW wind turbines with hub heights of 100

Table 2: The average capacity factors of existing VRE plants in each province of China in 2019[27]

Province Name	Capacity Factors of Wind	Capacity Factors of PV
Beijing	20.73%	15.13%
Tianjin	22.43%	13.12%
Hebei	24.47%	15.74%
Shanxi	21.89%	14.92%
Inner Mongolia	26.31%	18.98%
Liaoning	26.26%	15.95%
Jilin	25.30%	17.22%
Heilongjiang	26.52%	18.00%
Shanghai	23.57%	9.93%
Jiangsu	22.52%	13.61%
Zhejiang	23.86%	12.60%
Anhui	20.65%	12.58%
Fujian	30.13%	11.98%
Jiangxi	23.15%	12.15%
Shandong	21.27%	14.66%
Henan	16.89%	12.11%
Hubei	22.37%	12.73%
Hunan	22.37%	10.34%
Guangdong	18.40%	10.09%
Guangxi	27.23%	12.57%
Hainan	18.78%	12.16%
Chongqing	22.79%	6.93%
Sichuan	29.14%	17.81%
Guizhou	21.24%	12.47%
Yunnan	32.05%	15.45%
Tibet	24.81%	13.90%
Shaanxi	22.04%	14.54%
Gansu	20.40%	16.23%
Qinghai	19.90%	16.97%
Ningxia	20.67%	15.79%
Xinjiang	24.51%	16.27%

m that meet IEC standards. The wind energy resource data are provided by Vortex including wind speed, wind direction, air density, temperature, and other parameters. The wind data applied in this paper are in line with China’s reality.

In the ATB report, wind power resources are divided into 10 classes according to the average wind speed. The wind capacity factors over 45-50% reported by ATB are only for Class 1 - Class 3 whose potential wind plant capacities only account for 4% of the total[14]. The average wind capacity factor over the last three years in the US is 34.8% for the existing plants[24]. The number in China is 23.8%[15, 25]. The wind capacity factors in each province of China in 2019 are presented in Table 2. The differences in wind resource quality, wind turbine type and system accommodating ability together cause the differences in the average wind capacity factor according to our previous work[26]. Because the comparison of wind power resources between China and the US is not the focus of this paper, we will not discuss it in detail here.

The point is that the modelling of wind power in this paper is based on the practical situation of China’s power system. Hence, we think the setting on wind power in our model is reasonable. We are sorry that the data source is not clarified clearly in the previous version and caused a misunderstanding.

Comment 7: Similarly for solar, authors use a capacity factor of 15-18%, but ATB reported capacity factors are in excess of 25-30% due to inverter overloading etc - which would also explain somewhat higher capital cost in ATB.

Reply: Thank you for your careful review. We use the China-based capital costs, not the US-based costs, for PV power as mentioned in previous responses. The current average investment costs of PV power in the US are \$1300/kW according to ATB 2021[1]. However, the current average investment costs of PV power in China are about 4599 CNY/kW (\$666.8/kW)[2, 3]. Both the investment costs and capacity factors used in the model are China-based, which matches each other. The solar resource data are provided by SolarGIS including global horizontal irradiance and direct normal irradiance. The capacity factors are based on the generation characteristics of 300W monocrystalline PV modules with fixed-tilt systems, which are the main module type installed in PV plants in China. However, the capacity factors in ATB reports are for a one-axis tracking system[28]. This explains the higher capacity factors and capital costs in ATB. The inverter overloading may also be the reason.

The capacity factor of PV in China is around 15-18%. We investigated the PV capacity factors in each province of China in 2019 as presented in Table 2. The average PV capacity factor in China is 14.7% for the existing plants[15]. The PV data including the capital costs and capacity factors are in line with China’s reality in this paper. Hence, we think the setting on PV power in our model is reasonable. Again, we are sorry that the data source is not clarified clearly in the previous version and caused a misunderstanding.

Table 3: Mean absolute error between the trajectories of ATB 2020 and ATB 2021

Onshore wind	Offshore wind	PV utility	PV distributed	CSP	BESS	Nuclear	Biomass
10.14%	1.78%	4.29%	5.98%	3.56%	4.06%	1.16%	1.16%

Comment 8: Also, NREL has published ATB 2021, which shows even lower renewable prices.

Reply: Thank you for your insightful comment. We have updated the percentage trajectories applied in the GTEP model to ATB 2021. In addition, other parameters such as VRE capital costs have also been updated to the current status of China in 2020 based on the recently published reports [2, 3, 4]. The data used in the previous manuscript are based on the cost reports [15, 16] and ATB 2020 [14] as mentioned in the reply to Comment 4. The CNY/kW values for the investment costs of generation and transmission units are shown in the fifth column of Supplementary Table 14. We believe the updated input data are the most timely data we could access currently. Based on the updated data, we solved the planning problem again and recalculated the electricity supply costs and all other results.

For most of the generation technologies, the trajectories in the ATB 2021 are lower. The differences of the percentage trajectories between ATB 2020 and ATB 2021 under the moderate case are presented in Fig. 2. The mean absolute error of each generation technology is provided in Table 3. The mean absolute error between the trajectories of the two versions is 4.1%. For onshore wind power, the percentage trajectory of the 2021 version is about 10% points lower than that of the 2020 version, which is the largest difference. The differences between the two versions of projections are minor for other low-carbon generation and storage technologies.

There is no significant difference in the electricity supply cost and power system morphology results between the two versions. The major differences are presented as follows:

- The electricity supply costs decrease slightly with the updated input data. In the revised manuscript, the supply costs would increase by 9.6 CNY¢/kWh (1.39 USD¢/kWh). The comparison of the electricity supply costs under CN2050 scenarios between ATB 2020 and ATB 2021 is shown in Fig. 3. The average result differences are only 0.93 CNY¢/kWh (1.6% of the largest electricity costs).
- Because of the lower renewable prices, the total capacities of VRE increase from about 5.5 TW to 5.8 TW. To accommodate the additional VRE power, the installed capacities of ESSs increase by 99.2 GW.
- Meanwhile, the total investments in biomass power and bio-CCS are reduced by 47.34 GW since VRE accounts for more power generation. Consequently, the negative carbon emission contributed by bio-CCS units is reduced, which lead to the increase of marginal carbon price. The marginal price of carbon emission reaches 1444.2 CNY/t (209.4 USD/t) in 2050 with the updated input data.

In general, the changes in the input data have minor impacts on the quantitative results of the GTEP model. Most discussion results and policy implications keep the same. Please see the details of other results in the revised manuscript.

In addition, we have also investigated other capital cost projections of low-carbon generation and storage technologies from CRISO [29] and Wiser et al. [30]. The cost projection results in the NREL ATB report are relatively lower compared with the projection results mentioned above. Hence, the electricity supply costs to achieve carbon neutrality could be higher using the projections from CRISO [29] and Wiser et al. [30]. We understand that there is an inherent error between the forecast data and the actual situation. That is why we conducted the sensitivity analysis on several critical factors.

Figure 2: Differences of the percentage trajectories between ATB 2020 and ATB 2021 under the moderate case[14, 1]

Figure 3: Comparison between the electricity supply costs using ATB 2020 and ATB 2021

Comment 9: Points 4-7 make me question the study's incremental cost numbers and other results.

Reply: Thank you for your careful review. Points 4-7 are mainly derived from the misunderstanding that we used the US-based investment costs of clean technologies. All data used in this article are based on the practical situation of China's power system and China's renewable energy potential. It is our mistake that we did not make this clear in the main article. We have explained the related problems point-by-point in the responses to Points 4-7. Hope our replies can clear up the reviewer's doubts.

Comment 10: What are the total emissions from the power sector by 2050 and key intermediate years?

Reply: Thank you for your insightful comment. The annual carbon emission trajectories and specific data are presented in Fig. 4(a). The total carbon emissions from the beginning to the key years are presented in Fig. 4(b). Note that one planning stage covers five years in the GTEP model, as does the real scheme in China. The model constrains the annual carbon emission of the key years.

Figure 4: Annual and Total Carbon Emission in the key years

The total emissions by the key intermediate years shown in Fig. 4(b) are calculated using the trapezoidal rule where the carbon emissions in the intermediate years are calculated using linear interpolation. For example, the carbon emissions in 2020 and 2025 under the BAU scenario are 4.545 billion tonnes and 5.027 billion tonnes, respectively. Then the total emissions during the first five-year period is 23.930 billion tonnes calculated as follows:

$$E_{2021-2025} = \frac{E_{2020} + E_{2025}}{2} * 5 = \frac{4.545 + 5.027}{2} * 5 = 23.930 \text{ billion tonnes} \quad (4)$$

Comment 11: The discussion section only summarizes the key results that are already presented. There are no concrete conclusions drawn per se and no discussion on the policy directions needed.

Reply: Thank you for your insightful comment. We have rewritten the Discussion section and emphasized policy implications based on our research results. The first paragraph summarizes our findings on the power system morphology with the goals of carbon neutrality by 2050. The concrete conclusion about the power system morphology is that **wind and PV resources will serve as the core generating capacity to achieve carbon neutrality. Meanwhile, various flexible generation resources and transmission network expansion will serve as the auxiliary capacity to accommodate VRE and ensure system security.** Then, the other concrete conclusions on costs and policy implications are summarized as follows:

- **The current manufacturing capabilities of RE and ESS units are still insufficient to support the needed energy transformation** though the growth rate is already the highest in the world. Ensuring the integrity of the supply chain and implementing appropriate policy incentives are critical steps to achieve decarbonization of the power sector by 2050.
- The electricity supply cost increments not only derive from VRE capital costs but also the additional external costs to ensure system security due to intermittency and low inertia of VRE. Hence, **a more complete set of power market mechanisms including spot market, capacity market, and ancillary service market is urgently needed to translate the cost increments to the electricity price and allocate the benefits fairly.**
- Our analysis indicates that achieving carbon neutrality of the power sector in China by 2050 is technically feasible but increases the electricity supply costs significantly. The capital costs of RE and ESS units are the dominant factor affecting the costs of electricity supply according to the sensitivity analysis. **Policy subsidies based on feed-in tariffs or renewable energy certificates should be provided in the early stages of the newly-emerging low-carbon generation and storage technologies (like CSP and BESS) to spur technological innovation and scale effects ensuring the economics of transition.**

Please see the revised Discussion sections in the main paper for details.

Reviewer #2

This study comprehensively evaluate the cost increments to achieve the carbon neutrality in China electricity sector. This study use a bottom-up energy system model specialized for the capacity and transmission planning in the electricity sector. Even though the authors use lots of data and complex modelling framework, the manuscript is well-organized and well-written to follow easily. Most of analysis results make sense and several implications are expectable. However, I agree that the main contribution of this kind of study is to deliver the implications with quantitative and specific numbers.

Reply: Thank you for your insightful comments. We have tried our best to improve our manuscript based on your comments including a comparison with existing literature and an explanation of key parameters. Please see the responses below the comments and the revised manuscript for details.

In addition, please note that we have updated the future capital cost trajectories using ATB 2021[1] in the revised manuscript according to the suggestions in Comment 8 of Reviewer #1. Meanwhile, the parameters of electricity generation technologies have also been updated to the current status of China in 2020 based on the recently published reports [2, 3, 4]. We believe the updated input data are the most timely data we could access currently. We solved the planning problem again and recalculated the electricity supply costs and all other results based on the updated input data. The changes in the input data have minor impacts on the quantitative results of this article. Hence, most discussion results and policy implications keep the same. Please see the details in the reply to Comment 8 of Reviewer #1 and the revised manuscript.

Nevertheless, I think that authors can improve the manuscript based on my comments:

Comment 1: *The suggested model (GTEP model) seems to be a conventional power system planning model. If exist, please emphasize the new features of the model compared to previous studies.*

Reply: Thank you for your sensible and insightful comments. The main new features of the GTEP model in our paper compared with previous studies are threefold:

- **The supply curves of wind and PV power are integrated into the GTEP model in a piecewise manner to present the impacts of VRE resource spatial distribution.** Without considering the spatial distribution of LCOE, the cost will be underestimated by 2.2 CNY¢/kWh according to our study. The VRE supply curves of each province in China are shown in Supplementary Figure 7 and Supplementary Figure 8. Details on the piecewise calculation method are presented in Supplementary Note 2.
- **The model includes three kinds of power system security and stability constraints: power reserve limits, spinning reserve requirements, and minimum system inertia limits.** These three constraints respectively correspond to the security challenges of high RE penetrated power systems in the time scale of planning, operation, and transient stability. Such constraints determine the additional flexible resources that are required to accommodate the increasing RE. Their mathematical expressions are presented in Eq.(24) and Eq.(55)-Eq. (63) of SI. Few existing studies model minimum system inertia limits in the expansion model, which would underestimate the electricity supply costs.
- **We project local network expansion within provinces based on the historical data and the planning results of the GTEP model.** Meanwhile, we modify the local grid investment according to the changes in the capacity factors of the local capacity mix. The projection method is introduced in the Method section. The projection results for the local grids are shown in Supplementary Table 9 and Supplementary Table 10. The capital costs and maintenance costs of the transmission and distribution system within each province account for a considerable portion of the total electricity supply cost (about 15%), which is not considered in detail in previous studies.

Moreover, we have updated key parameters such as equipment costs, capacity mix, and grid structure to 2020 in the model. These data could be a useful reference for further research in the energy transition.

We compared our model with previous studies on the low-carbon transition of China in terms of other modelling details as shown in Table 4. In addition to the above three main features, our model provides advantages related to power equipment modelling, a high temporal resolution, and a long planning period duration. He et al. [9] proposed a “SwitchChina” model to study the impacts of rapid RE cost decrease on

low-carbon transition. The model has made progress in comprehensive national power system planning. But minimum system inertia limits, the critical challenges brought by high RE penetration are not considered. Due to its early publication time, some important new elements such as CCS and biomass energy did not participate in the low-carbon transition. To reduce the calculation burden, the minimum time resolution in “SwitchChina” model is set to six hours while the minimum time resolution is one hour in our model. This will impact the characterization of wind and PV intermittency during the operating simulation. Please see other model feature differences in Table 4.

It is hard to compare the cost results with existing articles directly because most studies have different target years and model settings. Some studies focus on the power system morphology and the results of electricity supply costs are not even discussed [11, 31]. The target years of [9] and [32] are 2030 and 2035, respectively. We pick two scenarios from [9] and [32] where the results are similar to ours for comparison as shown in Table 4. In [9], the electricity supply cost is 8.91 USD¢/kWh in 2030 with 80% carbon emission reduction. In [32], the electricity supply cost is 8.69 USD¢/kWh in 2035 with 75% RE penetration. It can be inferred that the electricity supply costs based on [9] and [32] will also be lower than that of our model under the scenario of carbon neutrality in 2050 because they require similar carbon emission targets to be reached earlier than our CN2050 scenario. The differences mainly come from the above three new features of our model mentioned above.

Chen et al. [12] proposed a single-period investment planning model for China’s power system where the RE penetration is forced in the target year of 2050. The RE penetration in [12] only includes wind, PV, and hydro power. The electricity supply cost is 2.72 USD¢/kWh under the base case with the RE penetration of 80%. The cost is much lower than our results (8.39 USD¢/kWh) in the CN2050 scenario whose RE penetration is 86.2%. We summarize the reasons for the lower costs in [12] as follows:

- Inner-provincial transmission network development and power losses caused by transmission are not considered in [12].
- Minimum system inertia limits are not considered in [12]. Moreover, the model does not set redundant power reserves as required by engineering practice (i.e. the power reserve rate rs in Eq. (24) of SI is set to 0).
- The VRE supply curves are not integrated into the model. In other words, the investment costs of all wind farms in each province are assumed to be the same in [12].
- The expansion model in [12] is single-period which means no investment decision or carbon emission limit in the key intermediate years is considered. Thus, the change process of RE unit capital costs and manufacturing capability limits are not modelled. The capital costs are calculated directly based on the value in the target year. However, our GTEP model is dynamic and multi-period.
- The LCOE of offshore wind power in [12] is too low, which is even lower than the onshore wind power. The setting is unexpected and difference with most predictions [14, 1, 29, 30].
- Carbon neutrality is not fully achieved in the 80% RE penetrated scenario of [12] since the capacity mix retains about 1000 GW coal power. The less stringent transition goal leads to lower electricity supply costs.

Hence, simply estimating electricity supply costs based on only the power balance or the single-period model will result in underestimation of power system decarbonization costs. In particular, a fuller cost accounting must take stock of important practical considerations by integrating VRE supply curves into models, considering operational security concerns, projecting developments of the local network, and high time resolution modelling over the planning periods. We have emphasized these new features of our model and added the comparison in Supplementary Note 6.

Table 4: Comparison with existing articles about the low-carbon transition in China

Model feature		This paper	[9]	[32]	[12]	[11]	[31]
Base year		2020	2015	2020	2019	2016	2018
Target year		2050	2030	2035	2050	2030	2030
Number of planning periods		6	3	1	1	14	1
Number of typical days		12	24	365	365	4	4
Minimum time resolution		one hour	six hours	one hour	one hour	one hour	one hour
Region of planning		Mainland China	Mainland China	Northwest China	Mainland China without Tibet	Mainland China	Mainland China
Generation	Onshore wind	Yes	Yes. But no distinction.	Yes	Yes	Yes. But no distinction.	Yes. But no distinction.
	Offshore wind	Yes		No	Yes		
	PV utility	Yes	Yes. But no distinction.	Yes. But no distinction.	Yes. But no distinction.	Yes. But no distinction.	Yes. But no distinction.
	PV distributed	Yes					
	CSP	Yes	No	No	No	No	No
	Hydro	Yes	Yes	Yes	Yes	Yes	No
	Nuclear	Yes	Yes	No	Yes	Yes	Yes
	Biomass	Yes	No	No	No	No	No
	Coal	Yes	Yes	Yes	Yes	Yes	Yes
	Gas	Yes	Yes	Yes	Yes	Yes	Yes
	Biomass-ccs	Yes	No	No	No	No	No
	Coal-ccs	Yes	No	No	Yes	No	No
Gas-ccs	Yes	No	No	No	No	No	
ESSs	Yes	Yes	Yes	Yes	Yes	No	Yes
Transmission	AC lines	Yes	Yes. But no distinction.	No	Yes	No	Yes. But no distinction.
	DC lines	Yes		No	Yes	No	
Are supply curves of VRE considered?		Yes	No	No	No	No	No
Are security constraints considered?	Power Reserve	Yes	Yes	Yes	Yes	No	No
	Spinning Reserve	Yes	Yes	Yes	Yes	No	No
	Minimum Inertia	Yes	No	No	No	No	No
Is the projection of the local grids expansion considered?		Yes	No	No	No	No	No
Electricity supply costs in the target year(USD¢/kWh)		8.39 (CN2050)	8.91 (C80)	8.69 (75% RE penetration)	2.72 (80% RE penetration)	N/A	N/A

*Note that Table 4 and Table 7 are the same. We present two same tables here to make the replies easier to read for reviewers.

Table 5: The parameter of generation technologies

Type	Start-up cost (10 ⁴ CNY/ MW)	Maintenance cost (10 ⁴ CNY/ MW/a)	Variable cost (10 ⁴ CNY/ MWh)	Investment cost (CNY/ kW)	Carbon emission (t/MWh)	Maximum ramp rate (%/10min)	Minimum output rate (%)	Lifetime (a)	Water consumption (t/MWh)	Inertia constant (s)
Hydro	0.00	25.93	0.0000	14561	0	100%	0%	50	0	2.83
PV utility	0.00	6.65	0.0000	4599	0	100%	0%	25	0	0
Onshore wind	0.00	14.60	0.0000	7600	0	100%	0%	25	0	0
Nuclear	2.20	55.64	0.0048	16000	0	0%	50%	50	4.167	4.07
Biomass	0.05	44.80	0.0506	10528	0.35	10%	35%	40	3.32	2.94
CSP	0.00	54.05	0.0000	27500	0	100%	10%	35	3.13	2.94
Coal	0.11	6.20	0.0274	4046	0.865	2%	40%	40	3.82	5.89
Coal CCS	0.11	24.40	0.0296	8009	0.865	2%	40%	40	5.02	5.89
Gas	0.03	9.80	0.0447	2387	0.312	7%	30%	40	0.97	4.97
Gas CCS	0.03	28.00	0.0499	6020	0.312	7%	30%	40	1.4	4.97
Bio CCS	0.05	44.80	0.0506	22319	0.35	10%	35%	40	4.007	2.94
PV distributed	0.00	6.30	0.0000	4599	0	100%	0%	25	0	0
Offshore wind	0.00	43.45	0.0000	17800	0	100%	0%	25	0	0

Comment 2: Even though this study focus on the costs in the electricity sector, the main manuscript and the supplementary notes do not provide the full information about the costs of electricity generation technologies. Please provide further information on this, including gas and CCS technologies.

Reply: Thank you for your sensible suggestions. The investment costs, maintenance costs, variable costs, and other technical parameters of generation units are presented in Supplementary Table 14. We copied the table into the response letter here which is Table 5. The investment costs of generation units are based on the authorized reports from China Electricity Council and China Electric Power Planning and Engineering Institute[2, 3]. The maintenance costs of RE refer to the report of International Renewable Energy Agency[4]. The maintenance costs of thermal power refer to [33].

We have considered the differences in the cost of fossil fuels in each province. The variable costs of thermal units including coal, gas, biomass, and CCS units for each province are present in Table 6 [33, 34, 35]. The variable costs of thermal units shown in Table 5 are the average of variable costs in each province weighted by their current installed capacities.

We applied the CCS unit model proposed in [35] where carbon capture can be dispatched. The carbon capture costs for Coal-CCS, Gas-CCS, and Bio-CCS are set to 390.8 CNY/tonne, 305.4 CNY/tonne, and 305.4 CNY/tonne respectively [36]. The detailed mathematical model is presented in Eq.(40)-Eq. (42) of SI. We have added the above information in the SI. Please see the revised SI for details.

Table 6: Variable costs of thermal units in different provinces (CNY/kWh)[33, 34, 35]

Province Name	Coal	Gas	Biomass	Coal-CCS	Gas-CCS	Bio-CCS
Beijing	0.264	0.487	0.570	0.327	0.524	0.623
Tianjin	0.264	0.487	0.570	0.327	0.524	0.623
Hebei	0.244	0.482	0.570	0.307	0.519	0.623
Shanxi	0.245	0.464	0.570	0.308	0.501	0.623
Inner Mongolia	0.213	0.324	0.570	0.276	0.362	0.623
Liaoning	0.276	0.482	0.533	0.339	0.519	0.583
Jilin	0.273	0.431	0.533	0.336	0.468	0.583
Heilongjiang	0.278	0.431	0.420	0.341	0.468	0.459
Shanghai	0.302	0.533	0.474	0.365	0.570	0.518
Jiangsu	0.290	0.528	0.531	0.353	0.565	0.581
Zhejiang	0.306	0.530	0.531	0.369	0.568	0.581
Anhui	0.284	0.510	0.531	0.347	0.547	0.581
Fujian	0.288	0.533	0.531	0.351	0.570	0.581
Jiangxi	0.307	0.477	0.542	0.370	0.514	0.592
Shandong	0.302	0.482	0.535	0.365	0.519	0.585
Henan	0.267	0.490	0.562	0.330	0.527	0.615
Hubei	0.319	0.477	0.414	0.382	0.514	0.453
Hunan	0.333	0.477	0.542	0.396	0.514	0.592
Guangdong	0.322	0.533	0.544	0.385	0.570	0.595
Guangxi	0.295	0.490	0.544	0.358	0.527	0.595
Hainan	0.320	0.401	0.544	0.383	0.438	0.595
Chongqing	0.303	0.401	0.387	0.367	0.438	0.423
Sichuan	0.321	0.403	0.513	0.384	0.440	0.561
Guizhou	0.252	0.419	0.568	0.315	0.456	0.621
Yunnan	0.303	0.419	0.448	0.366	0.456	0.489
Tibet	0.248	0.533	0.513	0.311	0.570	0.561
Shaanxi	0.248	0.324	0.508	0.311	0.362	0.556
Gansu	0.228	0.347	0.508	0.291	0.384	0.556
Qinghai	0.218	0.368	0.508	0.281	0.405	0.556
Ningxia	0.198	0.307	0.508	0.261	0.344	0.556
Xinjiang	0.176	0.276	0.508	0.239	0.313	0.556

Comment 3: Why the electricity demand are different among scenarios in Table 1? How to derive these?

Reply: Thank you for your careful review. The electricity demands are different among scenarios because the electrification levels of other industries are different in different emission reduction scenarios. The carbon reduction target is set for the entire energy system. For some industries that directly consume fossil energy, their carbon emission reduction goals are mainly achieved through electrification, which increases electric demands. For example, the development of electric vehicles increases the electric demands in the transportation field. The development of electric heating increases the electric demands in the heating industry. The power sector needs to bear larger electrification loads with more stringent carbon emission constraints.

This article focuses on the impacts of low-carbon goals on the power sector. Hence, the transition of other industries is beyond the scope of this article. We did not make detailed predictions on the electrification levels of various loads. The setting of electricity demands under different scenarios refers to the prediction from the work of the Institute of Climate Change and Sustainable Development[22]. We have explained this in the “Model and Scenario” section.

We understand that there is an inherent error between the projection data and the actual situation. In the sensitivity analysis, we studied the impact of load demand projection deviations on electricity supply costs. Load forecast deviation of 10% causes differences of 4.1 CNY¢/kWh in power supply cost at most as shown in the “Electricity supply costs sensitivities” section.

Comment 4: Please provide more comprehensive discussions about the sensitivity analysis

Reply: Thank you for your sensible suggestions. We added more comprehensive discussions in the “Electricity supply costs sensitivities” section. The discussions focus on the differences between the impact of various factors on the electricity supply costs, and the key factors to reduce costs. The added context is as follows:

“ Despite the influence of various uncertainties, increases in the electricity supply cost are almost inevitable. In most sensitivity analysis cases, the trajectory of cost changes is similar to that in the base case. In some cases, the costs slightly decline during the last period due to the maturity of generation technologies. These factors can be divided into two categories according to the periods in which they mainly affect the electricity supply costs. The RE and BESS capital costs, manufacturing capability, and security requirements most significantly impact the costs in the later stages of development, and the other factors impact the costs most significantly in the early to mid-term stages.

...

Although the electricity supply costs will inevitably increase, there are some ways to reduce the increase according to the above sensitivity analysis. The capital costs of RE and BESS units are the dominant factors affecting the electricity supply cost. Accelerating the cost decreasing by supporting the research and development of low-carbon generation and storage technologies would effectively reduce the transition costs associated with meeting the decarbonization target. In the early stage of the transition, it is critical to ensure a sufficient manufacturing capacity as much as possible. Moreover, reducing the total load demands by delaying the electrification process may reduce costs during this stage. However, this approach needs to be considered together with the low-carbon transition plans in other industries. In the later stage, a lack of interprovincial transmission capacity will lead to cost increases, and new transmission corridors will be needed. Reducing the reserve rates and minimum inertia limits also reduces the supply costs. However, this may not be allowed since ensuring system security is the first priority of power system planning and operation. The minimum value of the reserve rates and inertia limits to guarantee the system security stay unclear for future power systems, which requires further research on power system stability considering high renewable penetration. ”

Please see the revised main paper for more details.

Comment 5: Based on my experience, Figure 3(a) is called “capacity mix” and Figure 3(b) is called “generation mix” generally.

Reply: Thank you for your careful review. The captions of Figure 3(a) and Figure 3(b) have been changed to “capacity mix” and “generation mix” as suggested. Please see the revised paper for details.

Reviewer #3

Firstly, we would like to thank Reviewer #3 for the careful review throughout the entire manuscript. Typos and awkward English expressions are highlighted in the Review Attachment by Reviewer #3, which helped us improve the language greatly. We have corrected the writing mistakes and modified the inappropriate English use. The revised manuscript was also polished by “Nature Research Editing Service” and the native English speaker in our author group. Some highlighted sentences seem to have no language issues. Since no specific comments are given in the Review Attachment, we assume the reviewer has concerns or questions about the contents. Thus, necessary explanations are added in the replies of the Review Attachment directly. Please see the revised paper and the attachment for details. The responses to other specific comments are as follows.

In addition, please note that we have updated the future capital cost trajectories using ATB 2021[1] in the revised manuscript according to the suggestions in Comment 8 of Reviewer #1. Meanwhile, the parameters of electricity generation technologies have also been updated to the current status of China in 2020 based on the recently published reports [2, 3, 4]. We believe the updated input data are the most timely data we could access currently. We solved the planning problem again and recalculated the electricity supply costs and all other results based on the updated input data. The changes in the input data have minor impacts on the quantitative results of this article. Hence, most discussion results and policy implications keep the same. Please see the details in the reply to Comment 8 of Reviewer #1 and the revised manuscript.

Comment 1: Results are noteworthy and I think the work will be of significance to the field. The paper is similar He Gang et al "Rapid cost decrease..." among others. Consider providing a comparison with such papers, on both methods and results.

Reply: Thank you for your sensible and insightful comments. Other reviewers also proposed similar concerns. The main new features of the GTEP model in our paper compared with Prof. He Gang's work[9] and other similar articles are threefold:

- **The supply curves of wind and PV power are integrated into the GTEP model in a piecewise manner to present the impacts of VRE resource spatial distribution.** Without considering the spatial distribution of LCOE, the cost will be underestimated by 2.2 CNY¢/kWh according to our study. The VRE supply curves of each province in China are shown in Supplementary Figure 7 and Supplementary Figure 8. Details on the piecewise calculation method are presented in Supplementary Note 2.
- **The model includes three kinds of power system security and stability constraints: power reserve limits, spinning reserve requirements, and minimum system inertia limits.** These three constraints respectively correspond to the security challenges of high RE penetrated power systems in the time scale of planning, operation, and transient stability. Such constraints determine the additional flexible resources that are required to accommodate the increasing RE. Their mathematical expressions are presented in Eq.(24) and Eq.(55)-Eq. (63) of SI. Few existing studies model minimum system inertia limits in the expansion model, which would underestimate the electricity supply costs.
- **We project local network expansion within provinces based on the historical data and the planning results of the GTEP model.** Meanwhile, we modify the local grid investment according to the changes in the capacity factors of the local capacity mix. The projection method is introduced in the Method section. The projection results for the local grids are shown in Supplementary Table 9 and Supplementary Table 10. The capital costs and maintenance costs of the transmission and distribution system within each province account for a considerable portion of the total electricity supply cost (about 20%), which is not considered in detail in previous studies.

Moreover, we have updated key parameters such as equipment costs, capacity mix, and grid structure to 2020 in the model. These data could be a useful reference for further research in the energy transition.

We compared our model with previous studies on the low-carbon transition of China in terms of other modelling details as shown in Table 4. In addition to the above three main features, our model provides

advantages related to power equipment modelling, a high temporal resolution, and a long planning period duration. He et al. [9] proposed a “SwitchChina” model to study the impacts of rapid RE cost decrease on low-carbon transition. The model has made progress in comprehensive national power system planning. But minimum system inertia limits, the critical challenges brought by high RE penetration are not considered. Due to its early publication time, some important new elements such as CCS and biomass energy did not participate in the low-carbon transition. To reduce the calculation burden, the minimum time resolution in “SwitchChina” model is set to six hours while the minimum time resolution is one hour in our model. This will impact the characterization of wind and PV intermittency during the operating simulation. Please see other model feature differences in Table 4.

It is hard to compare the cost results with existing articles directly because most studies have different target years and model settings. Some studies focus on the power system morphology and the results of electricity supply costs are not even discussed [11, 31]. The target years of [9] and [32] are 2030 and 2035, respectively. We pick two scenarios from [9] and [32] where the results are similar to ours for comparison as shown in Table 4. In [9], the electricity supply cost is 8.91 USD¢/kWh in 2030 with 80% carbon emission reduction. In [32], the electricity supply cost is 8.69 USD¢/kWh in 2035 with 75% RE penetration. It can be inferred that the electricity supply costs based on [9] and [32] will also be lower than that of our model under the scenario of carbon neutrality in 2050 because they require similar carbon emission targets to be reached earlier than our CN2050 scenario. The differences mainly come from the above three new features of our model mentioned above.

Chen et al. [12] proposed a single-period investment planning model for China’s power system where the RE penetration is forced in the target year of 2050. The RE penetration in [12] only includes wind, PV, and hydro power. The electricity supply cost is 2.72 USD¢/kWh under the base case with the RE penetration of 80%. The cost is much lower than our results (8.39 USD¢/kWh) in the CN2050 scenario whose RE penetration is 86.2%. We summarize the reasons for the lower costs in [12] as follows:

- Inner-provincial transmission network development and power losses caused by transmission are not considered in [12].
- Minimum system inertia limits are not considered in [12]. Moreover, the model does not set redundant power reserves as required by engineering practice (i.e. the power reserve rate rs in Eq. (24) of SI is set to 0).
- The VRE supply curves are not integrated into the model. In other words, the investment costs of all wind farms in each province are assumed to be the same in [12].
- The expansion model in [12] is single-period which means no investment decision or carbon emission limit in the key intermediate years is considered. Thus, the change process of RE unit capital costs and manufacturing capability limits are not modelled. The capital costs are calculated directly based on the value in the target year. However, our GTEP model is dynamic and multi-period.
- The LCOE of offshore wind power in [12] is too low, which is even lower than the onshore wind power. The setting is unexpected and difference with most predictions [14, 1, 29, 30].
- Carbon neutrality is not fully achieved in the 80% RE penetrated scenario of [12] since the capacity mix retains about 1000 GW coal power. The less stringent transition goal leads to lower electricity supply costs.

Hence, simply estimating electricity supply costs based on only the power balance or the single-period model will result in underestimation of power system decarbonization costs. In particular, a fuller cost accounting must take stock of important practical considerations by integrating VRE supply curves into models, considering operational security concerns, projecting developments of the local network, and high time resolution modelling over the planning periods. We have emphasized these new features of our model and added the comparison in Supplementary Note 6.

Table 7: Comparison with existing articles about the low-carbon transition in China

Model feature		This paper	[9]	[32]	[12]	[11]	[31]
Base year		2020	2015	2020	2019	2016	2018
Target year		2050	2030	2035	2050	2030	2030
Number of planning periods		6	3	1	1	14	1
Number of typical days		12	24	365	365	4	4
Minimum time resolution		one hour	six hours	one hour	one hour	one hour	one hour
Region of planning		Mainland China	Mainland China	Northwest China	Mainland China without Tibet	Mainland China	Mainland China
Generation	Onshore wind	Yes	Yes. But no distinction.	Yes	Yes	Yes. But no distinction.	Yes. But no distinction.
	Offshore wind	Yes		No	Yes		
	PV utility	Yes	Yes. But no distinction.	Yes. But no distinction.	Yes. But no distinction.	Yes. But no distinction.	Yes. But no distinction.
	PV distributed	Yes					
	CSP	Yes	No	No	No	No	No
	Hydro	Yes	Yes	Yes	Yes	Yes	No
	Nuclear	Yes	Yes	No	Yes	Yes	Yes
	Biomass	Yes	No	No	No	No	No
	Coal	Yes	Yes	Yes	Yes	Yes	Yes
	Gas	Yes	Yes	Yes	Yes	Yes	Yes
	Biomass-ccs	Yes	No	No	No	No	No
	Coal-ccs	Yes	No	No	Yes	No	No
Gas-ccs	Yes	No	No	No	No	No	
ESSs	Yes	Yes	Yes	Yes	Yes	No	Yes
Transmission	AC lines	Yes	Yes. But no distinction.	No	Yes	No	Yes. But no distinction.
	DC lines	Yes		No	Yes	No	
Are supply curves of VRE considered?		Yes	No	No	No	No	No
Are security constraints considered?	Power Reserve	Yes	Yes	Yes	Yes	No	No
	Spinning Reserve	Yes	Yes	Yes	Yes	No	No
	Minimum Inertia	Yes	No	No	No	No	No
Is the projection of the local grids expansion considered?		Yes	No	No	No	No	No
Electricity supply costs in the target year(USD¢/kWh)		8.39 (CN2050)	8.91 (C80)	8.69 (75% RE penetration)	2.72 (80% RE penetration)	N/A	N/A

*Note that Table 4 and Table 7 are the same. We present two same tables here to make the replies easier to read for reviewers.

Comment 2: Your work supports the claims and conclusions you present but some assumptions like carbon reduction trajectory might be better if expressed explicitly in the body of the paper.

Reply: Thank you for your insightful comments. The annual carbon emission limit trajectories are presented in Fig. 5. The setting of annual carbon emission limit trajectories under different scenarios refers to the report from the Institute of Climate Change and Sustainable Development[22]. There is no carbon emission limit for the BAU scenario, so it is not shown in 5. We have put the figure in the body of the paper. We also added some other necessary assumptions in the text of the main paper. Please see the “Model and scenarios” section in the revised paper for details.

Figure 5: Annual carbon reduction trajectory under each scenario

The source data of all the figures in the main article were uploaded to a persistent repository for researchers [5]. The current link is now shared privately and its DOI is 10.6084/m9.figshare.16929340. The DOI will become active when the item is officially published.

Comment 3: Data analysis and discussion at times seem a bit disconnected, positioning discussion and tables/figures closer would be beneficial.

Reply: Thank you for your careful review. We have adjusted the position of tables/figures and discussion by changing the size or splitting the subgraphs. But we give priority to ensuring the figures pictures so that they can be seen clearly. Hence, discussion and tables/figures may still be not very close to each other at times. We hope the reviewer could understand this little flaw because the current manuscript is only a draft. If the paper could be accepted, the layout and format will be transformed to the style of Nature Communications by the professional journal staff after all. Please see the revised paper for details.

Comment 4: In addition, it seems that there is a bit of lack of clarity on capital/investment/overnight costs. Either explicitly say that they are interchangeable or define them and stick to one of the terms.

Reply: Thank you for your sensible suggestion. We have changed all the related words to capital costs in this paper. We clarified that capital costs are the one-time expenses incurred on the construction and manufacture of new electrical equipment including generation and transmission, which may be also called investment costs or overnight costs in the reference literature. Please see the caption of Fig.8 in the revised paper for details.

Comment 5: The methodology is sound and other papers use similar planning and optimization models. One detail I do not see being provided in the supplemental information is load estimates at the provincial levels for each of the scenarios.

Reply: Thank you for your sensible suggestion. The annual load demands at the provincial level in each of the scenarios are presented in Table 8-Table 10. The load demands in 2020 refer to [25]. The power sector needs to bear larger electrification loads under more stringent carbon emission scenarios. The annual future load demands under different scenarios refer to the prediction from the work of the Institute of Climate Change and Sustainable Development[22]. The load growth rate of each province refers to [37]. These data were added to the SI. Please see Supplementary Tables 21-23 in the revised SI for details.

Table 8: Annual load demands at the provincial level in the NDC/BAU scenario (TWh)

Province Name	2020	2025	2030	2035	2040	2045	2050
Beijing	119.07	128.29	145.23	156.90	164.72	172.25	179.47
Tianjin	93.88	101.15	114.51	123.70	129.87	135.81	141.50
Hebei	386.98	416.92	472.00	509.90	535.33	559.81	583.27
Shanxi	227.95	245.58	278.03	300.35	315.33	329.75	343.57
Inner Mongolia	365.77	394.08	446.14	481.96	506.00	529.14	551.31
Liaoning	241.20	251.22	278.88	295.42	305.58	314.83	323.19
Jilin	76.87	80.06	88.87	94.14	97.38	100.33	102.99
Heilongjiang	100.72	104.90	116.46	123.36	127.60	131.47	134.96
Shanghai	159.67	179.61	200.37	213.29	220.63	227.31	233.35
Jiangsu	688.10	774.04	863.50	919.21	950.83	979.63	1005.63
Zhejiang	514.38	578.62	645.50	687.15	710.78	732.31	751.75
Anhui	249.43	280.58	313.01	333.20	344.66	355.11	364.53
Fujian	246.26	277.02	309.04	328.98	340.29	350.60	359.91
Jiangxi	157.88	175.07	199.17	216.22	229.26	242.12	254.77
Shandong	655.27	705.97	799.24	863.41	906.48	947.92	987.64
Henan	333.97	370.33	421.31	457.37	484.95	512.16	538.92
Hubei	227.32	252.07	286.77	311.32	330.09	348.61	366.83
Hunan	190.55	211.29	240.38	260.96	276.69	292.22	307.49
Guangdong	702.67	790.43	921.46	1025.05	1086.86	1147.84	1207.81
Guangxi	199.81	224.76	262.02	291.48	309.06	326.40	343.45
Hainan	37.52	42.21	49.21	54.74	58.04	61.30	64.50
Chongqing	121.64	140.13	164.96	185.29	200.38	215.83	231.63
Sichuan	275.66	317.56	373.82	419.91	454.09	489.11	524.91
Guizhou	153.73	172.93	201.60	224.26	237.79	251.13	264.25
Yunnan	180.24	202.75	236.36	262.93	278.78	294.42	309.81
Tibet	10.60	12.21	14.38	16.15	17.46	18.81	20.19
Shaanxi	194.85	223.40	269.43	310.07	350.42	394.44	442.38
Gansu	129.88	148.91	179.59	206.68	233.57	262.92	294.87
Qinghai	74.22	85.09	102.62	118.10	133.47	150.24	168.50
Ningxia	108.67	124.60	150.27	172.94	195.44	219.99	246.73
Xinjiang	286.26	328.21	395.84	455.55	514.82	579.49	649.92

Table 9: Annual load demands at the provincial level in the GM2.0 scenario (TWh)

Province Name	2020	2025	2030	2035	2040	2045	2050
Beijing	119.07	130.67	147.06	160.21	170.88	181.21	191.14
Tianjin	93.88	103.03	115.95	126.32	134.73	142.87	150.71
Hebei	386.98	424.67	477.94	520.67	555.36	588.90	621.20
Shanxi	227.95	250.15	281.53	306.69	327.13	346.89	365.91
Inner Mongolia	365.77	401.40	451.75	492.14	524.93	556.64	587.17
Liaoning	241.20	255.89	282.39	301.65	317.01	331.20	344.21
Jilin	76.87	81.55	89.99	96.13	101.03	105.55	109.69
Heilongjiang	100.72	106.85	117.92	125.97	132.38	138.30	143.74
Shanghai	159.67	182.95	202.89	217.80	228.89	239.13	248.52
Jiangsu	688.10	788.42	874.37	938.62	986.40	1030.55	1071.04
Zhejiang	514.38	589.38	653.62	701.65	737.37	770.37	800.64
Anhui	249.43	285.80	316.95	340.24	357.56	373.56	388.24
Fujian	246.26	282.17	312.93	335.92	353.03	368.82	383.32
Jiangxi	157.88	178.33	201.68	220.79	237.84	254.71	271.34
Shandong	655.27	719.09	809.29	881.64	940.39	997.18	1051.88
Henan	333.97	377.21	426.61	467.03	503.10	538.78	573.98
Hubei	227.32	256.76	290.38	317.89	342.44	366.73	390.69
Hunan	190.55	215.22	243.41	266.47	287.05	307.40	327.49
Guangdong	702.67	805.12	933.06	1046.69	1127.53	1207.49	1286.37
Guangxi	199.81	228.94	265.32	297.63	320.62	343.36	365.79
Hainan	37.52	42.99	49.83	55.89	60.21	64.48	68.69
Chongqing	121.64	142.73	167.03	189.21	207.87	227.05	246.69
Sichuan	275.66	323.46	378.52	428.77	471.08	514.53	559.05
Guizhou	153.73	176.15	204.14	229.00	246.68	264.18	281.43
Yunnan	180.24	206.52	239.33	268.48	289.21	309.73	329.96
Tibet	10.60	12.44	14.56	16.49	18.12	19.79	21.50
Shaanxi	194.85	227.55	272.82	316.62	363.53	414.94	471.15
Gansu	129.88	151.68	181.85	211.05	242.31	276.58	314.05
Qinghai	74.22	86.67	103.92	120.60	138.46	158.05	179.46
Ningxia	108.67	126.91	152.16	176.59	202.75	231.43	262.77
Xinjiang	286.26	334.31	400.82	465.16	534.08	609.61	692.19

Table 10: Annual load demands at the provincial level in the CN2050 scenario (TWh)

Province Name	2020	2025	2030	2035	2040	2045	2050
Beijing	119.07	142.90	167.61	184.30	195.54	206.39	216.82
Tianjin	93.88	112.67	132.15	145.31	154.17	162.73	170.95
Hebei	386.98	464.41	544.73	598.96	635.49	670.75	704.66
Shanxi	227.95	273.56	320.87	352.81	374.33	395.10	415.08
Inner Mongolia	365.77	438.96	514.88	566.14	600.67	634.00	666.05
Liaoning	241.20	279.83	321.85	347.01	362.74	377.23	390.45
Jilin	76.87	89.18	102.57	110.59	115.60	120.21	124.43
Heilongjiang	100.72	116.85	134.40	144.91	151.48	157.52	163.05
Shanghai	159.67	200.07	231.24	250.55	261.91	272.36	281.91
Jiangsu	688.10	862.21	996.56	1079.76	1128.71	1173.77	1214.93
Zhejiang	514.38	644.53	744.97	807.17	843.76	877.44	908.21
Anhui	249.43	312.54	361.24	391.40	409.15	425.48	440.40
Fujian	246.26	308.58	356.66	386.44	403.96	420.08	434.81
Jiangxi	157.88	195.01	229.86	253.99	272.15	290.11	307.80
Shandong	655.27	786.38	922.39	1014.22	1076.06	1135.77	1193.20
Henan	333.97	412.51	486.23	537.26	575.68	613.66	651.09
Hubei	227.32	280.78	330.96	365.70	391.85	417.70	443.18
Hunan	190.55	235.36	277.42	306.54	328.46	350.13	371.48
Guangdong	702.67	880.47	1063.45	1204.08	1290.20	1375.31	1459.20
Guangxi	199.81	250.37	302.40	342.39	366.88	391.08	414.93
Hainan	37.52	47.02	56.79	64.30	68.90	73.44	77.92
Chongqing	121.64	156.09	190.37	217.66	237.86	258.60	279.83
Sichuan	275.66	353.73	431.42	493.25	539.04	586.04	634.16
Guizhou	153.73	192.63	232.66	263.43	282.27	300.89	319.25
Yunnan	180.24	225.84	272.78	308.85	330.94	352.77	374.29
Tibet	10.60	13.61	16.59	18.97	20.73	22.54	24.39
Shaanxi	194.85	248.85	310.95	364.23	415.98	472.61	534.45
Gansu	129.88	165.87	207.27	242.78	277.27	315.02	356.24
Qinghai	74.22	94.78	118.44	138.73	158.44	180.01	203.57
Ningxia	108.67	138.79	173.43	203.14	232.00	263.59	298.08
Xinjiang	286.26	365.60	456.83	535.11	611.13	694.33	785.18

References

- [1] National Renewable Energy Laboratory. 2021 annual technology baseline electricity data. <https://atb.nrel.gov/electricity/2021/data> (2021).
- [2] China Electricity Council. *Annual development report of China's power industry 2021* (in Chinese, China Architecture & Building Press, 2021).
- [3] China Electric Power Planning and Engineering Institute. *Report on China's Electric Power Development 2020* (in Chinese, People's Daily Press, Beijing, 2021).
- [4] IRENA. Renewable power generation costs in 2020. Tech. Rep., International Renewable Energy Agency (2021). URL <https://www.irena.org/publications/2021/Jun/Renewable-Power-Costs-in-2020>.
- [5] Zhuo, Z. Source data for "cost increase in the electricity supply to achieve carbon neutrality in china". <https://figshare.com/s/c64ef1fc6c71d13cf6ac> (2021).
- [6] Romero, R., Monticelli, A., Garcia, A. & Haffner, S. Test systems and mathematical models for transmission network expansion planning. *IEEE Proceedings-Generation, Transmission and Distribution* **149**, 27–36 (2002).
- [7] Faulstich, M. *et al.* Pathways towards a 100% renewable electricity system. Tech. Rep., German Advisory Council on the Environment (2011).
- [8] Child, M., Kemfert, C., Bogdanov, D. & Breyer, C. Flexible electricity generation, grid exchange and storage for the transition to a 100% renewable energy system in europe. *Renewable energy* **139**, 80–101 (2019).
- [9] He, G. *et al.* Rapid cost decrease of renewables and storage accelerates the decarbonization of china's power system. *Nature Communications* **11** (2020).
- [10] He, G. *et al.* Switch-china: a systems approach to decarbonizing china's power system. *Environmental science & technology* **50**, 5467–5473 (2016).
- [11] Chen, S., Liu, P. & Li, Z. Low carbon transition pathway of power sector with high penetration of renewable energy. *Renewable and Sustainable Energy Reviews* **130**, 109985 (2020).
- [12] Chen, X. *et al.* Pathway toward carbon-neutral electrical systems in china by mid-century with negative co2 abatement costs informed by high-resolution modeling. *Joule* **5**, 2715–2741 (2021). URL <https://www.sciencedirect.com/science/article/pii/S2542435121004505>.
- [13] Zhang, Q. & Chen, W. Modeling china's interprovincial electricity transmission under low carbon transition. *Applied Energy* **279**, 115571 (2020).
- [14] National Renewable Energy Laboratory. 2020 annual technology baseline electricity data. <https://www.nrel.gov/news/program/2020/2020-annual-technology-baseline-electricity-data-now-available.html> (2020).
- [15] China Electricity Council. *Annual development report of China's power industry 2020* (in Chinese, China Architecture & Building Press, 2020).
- [16] China Electric Power Planning and Engineering Institute. *Report on China's Electric Power Development 2019* (in Chinese, China Electric Power Press, Beijing, 2020).
- [17] Wiser, R. H. *et al.* Wind energy technology data update: 2020 edition. Tech. Rep., Lawrence Berkeley National Laboratory (2020). URL <https://emp.lbl.gov/wind-technologies-market-report>.
- [18] Bolinger, M., Seel, J., Robson, D. & Warner, C. Utility-scale solar data update: 2020 edition. Tech. Rep., Lawrence Berkeley National Laboratory (2020). URL <https://www.osti.gov/biblio/1706670>.

- [19] U.S. Energy Information Administration. U.s. imports of solar photovoltaic modules mainly come from asia. <https://www.eia.gov/todayinenergy/detail.php?id=34952> (2018).
- [20] Muñoz, F. D., Suazo-Martínez, C., Pereira, E. & Moreno, R. Electricity market design for low-carbon and flexible systems: Room for improvement in chile. *Energy Policy* **148**, 111997 (2021).
- [21] Falchetta, G., Hafner, M. & Tagliapietra, S. Pathways to 100% electrification in east africa by 2030. *The Energy Journal* **41** (2020).
- [22] Institute of Climate Change and Sustainable Development. *China's long-term low-carbon development strategy and pathway* (Springer, Singapore, 2021).
- [23] National Renewable Energy Laboratory, U., Oak Ridge National Laboratory. Annual technology baseline the 2021 electricity update. <https://www.nrel.gov/docs/fy21osti/80095.pdf> (2021).
- [24] U.S. Energy Information Administration. Capacity factors for utility scale generators primarily using non-fossil fuels. https://www.eia.gov/electricity/monthly/epm_table_grapher.php?t=epmt_6_07_b (2021).
- [25] China Electricity Council. National electric power industry statistics express 2020. Tech. Rep., in Chinese, Power Statistics and Data Center, China Electricity Council (2021).
- [26] Lu, X. *et al.* Challenges faced by china compared with the us in developing wind power. *Nature Energy* **1** (2016).
- [27] State Grid Energy Research Institute. *2020 China Generation Development Analysis Report* (in Chinese, China Electric Power Press, Beijing, 2020).
- [28] National Renewable Energy Laboratory. 2021 electricity annual technology baseline: Utility-scale pv. https://atb.nrel.gov/electricity/2021/utility-scale_pv#MIISNV9Q (2021).
- [29] Graham, Paul and Hayward, Jenny and Foster, James and Havas, Lisa. Gencost project data. <https://doi.org/10.25919/rpwh-wc51> (2021).
- [30] Wisser, R. *et al.* Expert elicitation survey predicts 37% to 49% declines in wind energy costs by 2050. *Nature Energy* **6**, 555–565 (2021).
- [31] Sharifzadeh, M., Hien, R. K. T. & Shah, N. China's roadmap to low-carbon electricity and water: Disentangling greenhouse gas (ghg) emissions from electricity-water nexus via renewable wind and solar power generation, and carbon capture and storage. *Applied Energy* **235**, 31–42 (2019).
- [32] Chen, X. *et al.* Power system capacity expansion under higher penetration of renewables considering flexibility constraints and low carbon policies. *IEEE Transactions on Power Systems* **33**, 6240–6253 (2018).
- [33] Zhang, D. & Paltsev, S. The future of natural gas in china: Effects of pricing reform and climate policy. *Climate Change Economics* **7**, 1650012 (2016).
- [34] National Energy Administration. 2018 national electricity price regulatory bulletin. Tech. Rep. (2019). URL , inChinese, http://www.nea.gov.cn/138530255_15729388881531n.pdf.
- [35] Ji, Z. *et al.* Low-carbon power system dispatch incorporating carbon capture power plants. *IEEE Transactions on Power Systems* **28**, 4615–4623 (2013).
- [36] Irlam, L. Global costs of carbon capture and storage. *Global CCS institute* (2017).
- [37] GEIDCO. Research Report on China's "14th Five-Year" Power Development Plan. Tech. Rep., in Chinese, Global Energy Interconnection Development and Cooperation Organization (2020).

Reviewer comments, second round review –

Reviewer #1 (Remarks to the Author):

The paper presents a detailed planning model for evaluating the increase in electricity costs required for a full decarbonization of China. The methodology focuses only on the power sector, accounting for the electricity consumption increase from electrification of other energy sectors. This is a wide-ranging study, that leverages the best (nation-wise and internationally-wise) data, with a detailed granularity from all relevant actors in the field. The main contribution of the work is to provide accurate quantitative values for future costs, along with the implications for the wide economy. Even though the study is focused on China, the methodological contributions in the representation of the different technologies and network considerations provide highly valuable insights for all policy makers worldwide. Overall, the paper is very well written and structured. The reviewer has the following comments to further clarify the contributions and conclusions of the work.

-In page 4, it is mentioned "Such detailed modelling of RE development costs, power system security constraints, and transmission line investments ensures an accurate assessment of electricity supply costs with the carbon neutrality target." It is unclear why the authors refer only to transmission line investment since the fourth above-mentioned assumption states that the investment in regional networks (which I guess refer to distribution networks) are also considered (according to GTEP simulation results and historical network data). In light of the information provided in the section 'Local grid development projection', it may be valuable to explain sooner in the manuscript that distribution investment costs are not endogenously included in the model, but that they are integrated in an ex-post fashion.

-It is unclear what are the assumptions regarding the operation of the distribution level, (where there is a lot of potential of flexibility from load management strategies at both industrial and consumer levels), which may account for a non-negligible part of the total costs of the system. It may be valuable to explicitly explain the assumptions used regarding the potential from distributed assets such as electric vehicles and heat pumps?

-In page 10, line 192, the authors state that "the (marginal) 2020 carbon price in China's carbon market is only 49 CNY/t (7.1 USD/t) ». It is unclear what is the reference case. Since the marginal carbon price correspond to the variation of the total costs in response to a reduction of carbon emission, what is the zero carbon emission reference case? How does it differ from business-as-usual?

-What are the assumptions behind the development (towards 2050) of industrial and residential loads? What are the main drivers underlying their future evolution.

-Are there constraints for the minimum and maximum amount of VRE capacity that can be installed per year? How are they quantified?

Reviewer #2 (Remarks to the Author):

I agree that authors did their best to improve the manuscript. However, the new features of GTEP

model the authors summarized in the response for comment #1 are not new methodologically. Even the commercial models, like TIMES, have the features. Of course, I agree that this kind of study has the value for policy implication rather than the methodological contributions.

Reviewer #3 (Remarks to the Author):

This work presents noteworthy results as they provide other researchers in this area with solid assumptions in their work (economic, policy, trade). It builds on existing literature and adds further details beyond it. In particular, it provides researchers with estimates for carbon transition costs for China, which might not be easily translatable to other regions, it might still provide background on methodology and estimates beyond China. I am not familiar with the GREAN model, but it seems similar to other electricity expansion planning models that use physical constraints and resources, as well as forecasted prices in their decision making. Although not all data or models are shared in the paper, the authors open the door for supplying data not easily found on the literature (which is often the case for China) and models used in the paper to reasonable requests.

Response to Reviewers' Comments on “Cost Increase in the Electricity Supply to Achieve Carbon Neutrality in China”

March 10, 2022

First of all, we would like to thank the reviewers for the time and effort that you have put into reviewing the previous version of the manuscript. Thank you all very much for your careful review and valuable comments. The suggestions and comments have enabled us to further improve the quality of our paper.

Our responses to the comments and questions are given directly below them in this letter. We have tried our best to improve the presentation of the paper. All the modifications in the article have been **highlighted in grey**. This round of revisions mainly added necessary explanations of various assumptions in the manuscript. The major improvements and responses are summarized as follows:

- The evaluation method of distribution network developments is clarified. The costs brought about by distribution grid developments are considered in an ex-post fashion and not endogenously optimized in the GTEP model. They are projected based on the GTEP simulation results and historical network data.
- The assumptions regarding the operation of the distribution level are explained. Particularly, the setting and modeling method of load management strategies are further discussed.
- We emphasize the calculation methods of the marginal carbon price and explain why its calculation does not require a reference case.
- The data source and the reference for parameter settings including load demands and manufacturing capabilities are provided. We illustrate that the focus of this paper is the impacts on power system morphology and costs under carbon neutrality. The specific developments of load components are beyond the scope of this paper.
- We clarify the major contributions of this paper in the response letter. This paper provided quantified conclusions and policy implications on carbon neutrality in China's power system and established the dataset for future work about China's energy transition. Methodologically, the GTEP model considered the new features of the high renewable energy penetrated power systems, and an evaluation framework of electricity supply costs which projects the local grid developments in an ex-post fashion is proposed.

Please see the revised manuscripts and the following responses for details.

Response to the Reviewers

Reviewer #1

The paper presents a detailed planning model for evaluating the increase in electricity costs required for a full decarbonization of China. The methodology focuses only on the power sector, accounting for the electricity consumption increase from electrification of other energy sectors. This is a wide-ranging study, that leverages the best (nation-wise and internationally-wise) data, with a detailed granularity from all relevant actors in the field. The main contribution of the work is to provide accurate quantitative values for future costs, along with the implications for the wide economy. Even though the study is focused on China, the methodological contributions in the representation of the different technologies and network considerations provide highly valuable insights for all policy makers worldwide. Overall, the paper is very well written and structured. The reviewer has the following comments to further clarify the contributions and conclusions of the work.

Reply: Thank you for acknowledging the contributions of our work and giving insightful comments. We have improved the manuscript according to the comments and have provided further explanations about the assumptions. Hope our replies can clear up your concerns. Please see the following responses below the comments and the revised manuscript for details.

Comment 1: In page 4, it is mentioned "Such detailed modelling of RE development costs, power system security constraints, and transmission line investments ensures an accurate assessment of electricity supply costs with the carbon neutrality target." It is unclear why the authors refer only to transmission line investment since the fourth above-mentioned assumption states that the investment in regional networks (which I guess refer to distribution networks) are also considered (according to GTEP simulation results and historical network data). In light of the information provided in the section 'Local grid development projection', it may be valuable to explain sooner in the manuscript that distribution investment costs are not endogenously included in the model, but that they are integrated in an ex-post fashion.

Reply: Thank you for your detailed review and sensible suggestions. Your understanding of distribution network modeling in this article is correct. The costs brought about by distribution grid developments are considered in an ex-post fashion and not endogenously optimized in the GTEP model. As explained in the "Local grid development projection" section, only the expansion of inter-provincial transmission lines is modeled in the GTEP model. Local grid planning (including within-provincial transmission lines and distribution networks) is not directly optimized in the GTEP model because of the inaccessibility of data and calculation issues. Hence, we project the investment of local grid expansion based on the historical local network capacity and the planning results of the GTEP model. The projection details and results are also shown in the "Local grid development projection" section and the supplementary information(SI).

Thus, the original statement in Page 4 was inappropriate and was changed to "*Such detailed modeling of RE development costs, power system security constraints, and **grid expansion investments** ensures an accurate assessment of electricity supply costs with the carbon neutrality target.*" Moreover, we further explained this modeling method in Page 2 and Page 4 as suggested. In Page 2, Line 41, we added that "*For the local grids including within-provincial transmission network and distribution networks, we project expansion based on historical network data and simulation results.*" In Page 4, Line 82, we added that "*The local network expansion including within-provincial transmission and distribution networks is projected according to GTEP simulation results and historical network data in an ex-post fashion (see the Methods section)*" Please see the revised manuscript for details.

Comment 2: It is unclear what are the assumptions regarding the operation of the distribution level, (where there is a lot of potential of flexibility from load management strategies at both industrial and consumer levels), which may account for a non-negligible part of the total costs of the system. It may be valuable to explicitly explain the assumptions used regarding the potential from distributed assets such as electric vehicles and heat pumps?

Reply: Thank you for your insightful comments. In the GTEP model, the load management strategies are modeled as load shedding. The cost of load shedding is set to 3 CNY/kWh according to the Annual

Development Report of China's Power Industry 2021 [1]. At present, load management strategies (also called demand responses) are only carried out in a few pilots in China. In Tianjin city, the current average price for demand response is approximately 3.3 CNY/kWh. In Jiangsu province, the prices for demand response vary between 1.33 CNY/kWh and 5 CNY/kWh. There is a lot of potential of flexibility from load management strategies but it is currently expensive. The lowest demand response price (1.33 CNY/kWh) is even much larger than the highest Levelized Cost of Electricity (LCOE) of RE generation which is 0.73 CNY/kWh for CSP units. Hence, demand response is normally the last option for power system operation and modelling it as load shedding is sensible. Load management strategies are not active according to the GTEP results in our model. At present, the actual load management strategies in China are mainly applied as reserves for low-probability extreme scenarios, such as disaster accidents and peak loads, rather than conventional dispatch means. We believe the results where the load management strategies are not active are reasonable since the GTEP model contains operation simulation for typical days rather than low-probability extreme scenarios.

The distributed power generators in the distribution network, such as distributed PV and distributed biomass units, are aggregated as one unit for each province. It is assumed that they could be dispatched in a centralized manner through aggregators. Their operating costs are the same as other generators. We have emphasized the assumptions regarding the operation of the distribution level including load management strategies and distributed power generators in Supplementary Note 1.1. Please see the revised manuscript for details.

The specific load components are not precisely modelled in the GTEP model. For the controllable loads such as electric vehicles and heat pumps, their equivalent energy storage functions and flexibility are uniformly modelled as the aggregated energy storage devices and demand responses like the distributed power generators. As the new loads from the electrification, electric vehicles and heat pumps are considered in the total load demands but not individually modelled in the GTEP model. The electrification of the transportation sector and the heating sector through electric vehicles and heat pumps will inevitably lead to an increase in the load. Under different carbon emission reduction targets, the electrification level of loads will be different, which is reflected in the different load demand settings for different scenarios as shown in Table 1. The setting of electricity demands under different scenarios refers to the prediction from the work of the Institute of Climate Change and Sustainable Development [2]. This article focuses on the impacts of low-carbon goals on the power sector. Hence, the transition of other industries is beyond the scope of this article. We did not make detailed predictions on the electrification levels of various loads.

Comment 3: In page 10, line 192, the authors state that "the (marginal) 2020 carbon price in China's carbon market is only 49 CNY/t (7.1 USD/t). It is unclear what is the reference case. Since the marginal carbon price correspond to the variation of the total costs in response to a reduction of carbon emission, what is the zero carbon emission reference case? How does it differ from business-as-usual?"

Reply: Thank you for your careful review. Calculating the marginal carbon prices does not need a reference case. As explained in Supplementary Note 3, marginal carbon prices are the cost increase per tonne of carbon emission reduction under certain emission limits. They represent the sensitivity of the total costs to carbon emission limits. From the perspective of an optimization problem, the marginal carbon prices are the shadow prices of the carbon emission limit constraints (Eq. (23) in Supplementary Note 1). Shadow prices reflect the scarcity of related resources [3], which are the carbon emission budget in this paper. There are alternative names for shadow prices, such as optimal dual variable values or optimal Lagrange multipliers. In terms of definitions, marginal carbon prices are similar to the local marginal prices (LMP) of electricity which are also the shadow prices and have been widely used in the electricity market clearing. Hence, calculating the marginal carbon prices does not need a reference case. By solving the GTEP model, a linear programming problem, the shadow prices can be obtained directly because they are necessary intermediate parameters during the barrier algorithm [4]. The marginal carbon price in the BAU scenario is zero because there are no carbon emission constraints considered.

The number of 49 CNY/t in Page 10, Line 192 is the actual carbon market clearing price in 2020 China's carbon market. It is the average value over the 2020 [5]. This actual carbon market-clearing price is generated through bidding and market clearing. Therefore, there is no reference case for the price either. The current carbon market in China is for the whole society economic not only the power sector. 49 CNY/t

reflects the value of carbon emission budgets in 2020 over the whole society. As a benchmark, we compared it with the marginal carbon prices from the GTEP model in the paper to illustrate that the value of carbon emission budgets will soar under the carbon neutrality scenario. We further emphasized these in the “Costs of carbon neutrality” section and Supplementary Note 3.

Additionally, we also present the average carbon mitigation costs in the ‘Costs of carbon neutrality’ section. The calculation of the average carbon mitigation costs requires a reference case. We think this might be what the reviewer expected in the comment. The average carbon mitigation cost is the additional cost per tonne of carbon emission between two scenarios as described in Supplementary Note 3.3. Its value is numerically equal to the ratio of the difference between the total cost of the two scenarios and the difference between the carbon emission budget. Hence, the scenario with the larger carbon emission budget could be regarded as the reference case. We presented the average carbon mitigation costs between the four scenarios in Figure 7 of the main paper. The number between each pair of curves denotes the average carbon mitigation costs. For example, the average carbon mitigation cost is 699.6 CNY/t between GM2.0 and CN2050 as illustrated in Figure 7. Thus, the GM2.0 scenario is the reference case here. The average carbon mitigation cost is 43.1 CNY/t between BAU and NDC. Then, the BAU scenario is the reference case. Please see the “Costs of carbon neutrality” section and Supplementary Note 3.3 for details.

Comment 4: What are the assumptions behind the development (towards 2050) of industrial and residential loads? What are the main drivers underlying their future evolution.

Reply: Thank you for your sensible comments. As mentioned in the reply to Comment 2, we did not make detailed predictions on the various load component developments. The setting of load demands under different scenarios refers to the prediction from the work of the Institute of Climate Change and Sustainable Development [2]. The load demands are modelled in GTEP as input boundary conditions. Based on Ref. [2] and the information obtained by consulting its authors, the main drivers underlying the future load evolution are economic growth and electrification requirements. The electrification of industrial loads will mainly come from industrial electric boilers, metallurgical electric furnaces, and auxiliary electric motors. The electrification of residential loads will mainly come from air conditioning loads, electric heating, and electric vehicles. The industrial load will be the main part of the load demands, accounting for more than 56% of the total load. We have explained this in the “Model and Scenario” section.

Comment 5: Are there constraints for the minimum and maximum amount of VRE capacity that can be installed per year? How are they quantified?

Reply: Thank you for your detailed review. There are no constraints for the minimum amount of VRE capacity that can be installed per year, which means the newly installed VRE capacities could be zero in the GTEP model. However, there are constraints for the maximum VRE capacities that are newly installed per year because of the limits of manufacturing capability. The maximum VRE manufacturing capability is quantified based on historical data and expert consultations. The average growth capacities of wind and PV between 2016-2020 are 30.0 GW and 41.8 GW annually [6]. It is reported that approximately 100 GW VRE units have started construction at the end of 2021 [7]. Hence, we set the maximum VRE manufacturing capability in the first planning stage (2021-2025) to 100 GW per year for the base case. We consulted several experts from the China Electric Power Planning and Engineering Institute and set the future maximum VRE manufacturing capability at about 260 GW per year. We assumed that VRE manufacturing capability grows linearly and reaches a maximum around 2035. We explain this in the Supplementary Note 4. Please see the revised manuscript and Supplementary Figure 3(a) for the specific settings.

The actual manufacturing capacities in China in the future are highly uncertain, which mainly depends on the strength of policy support. Therefore, we made a sensitivity analysis of electricity supply costs on RE and BESS manufacturing capacities as shown in the “Electricity supply costs sensitivities” section. The results show that manufacturing capacities mainly affect the electricity supply costs in the mid-term stages of system development, and have less impact on the final electricity supply cost as presented in Figure 9 of the main article. This means even if the actual installed capacity growth rate is greater than the maximum manufacturing capacities we set, the final electricity supply cost will be close to our results. However, when the manufacturing capacity is too low to produce enough VRE units more than a certain threshold over the future 30 years (approximately 5000 GW according to our results), considerable investment in coal-based

CCS units would need to meet the emission reduction goals, leading to gross increases in costs (see the outliers in Figure 9 for the RE unit manufacturing capability). Hence, sufficient manufacturing capabilities are important for achieving the carbon neutrality transformation.

Reviewer #2

Comment 1: I agree that authors did their best to improve the manuscript. However, the new features of GTEP model the authors summarized in the response for comment #1 are not new methodologically. Even the commercial models, like TIMES, have the features. Of course, I agree that this kind of study has the value for policy implication rather than the methodological contributions.

Reply: Thank you for acknowledging the contributions of our work and giving insightful comments. We agree with you that the quantified conclusion and corresponding policy implications are critical contributions of this paper. Moreover, researchers can carry out further research on carbon neutrality in China's entire energy system based on the data provided in this paper which we believe are the most timely data we could access currently. We would like to clarify some potential methodological advantages of our work compared with commercial models, like TIMES:

- TIMES model is oriented to the development of the entire energy systems and does cover the basic elements of power system planning. However, the minimum inertia constraints and spinning reserve constraints are not considered according to the Documentation for the TIMES Model [8, 9] which are important features of the high renewable energy penetrated power systems. In the results of our paper, these secure constraints are important reasons for the investments of ESSs and other flexible units. Besides, some novel RE technologies such as CSP are not involved in the TIMES model. The GTEP model in this paper takes these factors into account.
- We project local network expansion within provinces based on the historical data and the planning results of the GTEP model in an ex-post fashion. The capital costs and maintenance costs of the transmission and distribution system within each province account for a considerable portion of the total electricity supply cost (about 15%). The evaluation of local network expansion is beyond the optimization model which is not considered in the TIMES model or previous studies as far as we know.

We believe the above modeling methods and evaluation framework could be implemented in TIMES by necessary extensions. However, these new features do not originally exist in the TIMES model and no previous studies have considered them based on the TIMES model or other commercial models as far as we know. Given necessary data in the corresponding regions, these innovative modeling methods can be applied to the studies of carbon neutrality costs and morphology in other regions.

Reviewer #3

Comment 1: This work presents noteworthy results as they provide other researchers in this area with solid assumptions in their work (economic, policy, trade). It builds on existing literature and adds further details beyond it. In particular, it provides researchers with estimates for carbon transition costs for China, which might not be easily translatable to other regions, it might still provide background on methodology and estimates beyond China. I am not familiar with the GREAN model, but it seems similar to other electricity expansion planning models that use physical constraints and resources, as well as forecasted prices in their decision making. Although not all data or models are shared in the paper, the authors open the door for supplying data not easily found on the literature (which is often the case for China) and models used in the paper to reasonable requests.

Reply: Thank you for acknowledging the contributions of our work and giving insightful comments. The GREAN model provides the potential and the supply curves of Wind and PV in each province for the GTEP optimization model. As mentioned in the comment, the contributions of this paper are three-fold:

- This paper provided quantified conclusions and corresponding policy implications on carbon neutrality in China's power system.
- The necessary dataset for the long-term development of China's power system is established and could be applied in future work about China's energy transition.
- This paper provides the GTEP model where new features of the high renewable energy penetrated power systems are modeled and the evaluation framework of electricity supply costs where the local grid developments are considered.

We would also like to thank this reviewer for understanding the data availability. The source data underlying all the figures above are provided as a Source Data file [10]. All data used for this study are available from cited publicly available sources or the authors upon reasonable request.

References

- [1] China Electricity Council. *Annual development report of China's power industry 2021* (in Chinese, China Architecture & Building Press, 2021).
- [2] Institute of Climate Change and Sustainable Development. *China's long-term low-carbon development strategy and pathway* (Springer, Singapore, 2021).
- [3] Heckman, J. Shadow prices, market wages, and labor supply. *Econometrica: journal of the econometric society* 679–694 (1974).
- [4] Conn, A., Gould, N. & Toint, P. A globally convergent lagrangian barrier algorithm for optimization with general inequality constraints and simple bounds. *Mathematics of Computation* **66**, 261–288 (1997).
- [5] Slater, H., de Boer, D., Qian, G. & Wang, S. 2020 china carbon pricing survey. Tech. Rep., China Carbon Forum (2020).
- [6] China Electric Power Planning and Engineering Institute. *Report on China's Electric Power Development 2020* (in Chinese, People's Daily Press, Beijing, 2021).
- [7] Xinhua News Agency. Large-scale wind and pv base projects in deserts, gobi and desert areas in my country started in an orderly manner. in Chinese, <https://news.un.org/en/story/2020/09/1073052> (2021).
- [8] Loulou, R., Goldstein, G., Kanudia, A., Lettila, A. & Remme, U. Documentation for the times model part i. Tech. Rep. (2021). URL https://iea-etsap.org/docs/Documentation_for_the_TIMES_Model-Part-I.pdf.
- [9] Loulou, R., Lehtila, A., Kanudia, A., Remme, U. & Goldstein, G. Documentation for the times model part ii. Tech. Rep. (2020). URL https://iea-etsap.org/docs/Documentation_for_the_TIMES_Model-Part-II.pdf.
- [10] Zhuo, Z. Source data for "cost increase in the electricity supply to achieve carbon neutrality in china". <https://figshare.com/s/c64ef1fc6c71d13cf6ac> (2021).

Reviewer comments, third round review –

Reviewer #1 (Remarks to the Author):

I thank the authors for their careful and complete response to my comments.
Overall, the authors are to be congratulated for the quality of the work. The considerable amount of efforts is tangible, and has led to a high-quality paper. I have no further comments.

Response to Reviewers' Comments on “Cost Increase in the Electricity Supply to Achieve Carbon Neutrality in China”

April 5, 2022

First of all, we would like to thank reviewers again for the time and effort that you have put into reviewing the previous version of the manuscript. Thank you all very much for your careful review and valuable comments. The suggestions and comments have enabled us to further improve the quality of our paper.

This round of revisions mainly modified the manuscript format to comply with the editorial requests and prepared the final submission. Our responses to the comments and requirements are given directly below them in this letter and the attached checklists. We believe the current version addresses the remaining concerns of the reviewers and the editorial requests. All the modifications in the article have been highlighted in grey. The major adjustments and responses are summarized as follows:

- The formats of the main text and Supplementary Information have been modified as required by the author checklist including the layout of figures, the manuscript structure, the complete information of the reference list, equation format requirements, the public data repository and so on. Please see the revised manuscript and the author checklist with responses for details.
- An updated reporting summary has been completed and uploaded as a supplementary information file with the revised manuscript named “nr-reporting-summary-updated.pdf”. All points on the reporting summary are addressed. Please see the responses to the comments in the checklist named “nr-reporting-summary-withReplies.pdf” for details.
- A separate point-by-point response to the reviewers' comments is provided in the submission.
- We confirmed the relevant funding awarded to authors and the author lists in the manuscript. Ning Zhang and Chongqing Kang are the corresponding authors of the article. Zhenyu Zhuo is the first author of the article.
- As required in the author checklist, we provide a brief summary of the main findings of the paper here as follows: *“This study indicates that approximately 5.8 TW of wind and solar photovoltaic capacity would be required to achieve carbon neutrality in China’s power system by 2050. The electricity supply costs would increase by 19.9% or 9.6 CNY¢/kWh.”*(236 characters including space)
- If it is possible, we would like to include the Twitter handles of our universities in the potential twitters about this work, which are @Tsinghua.Uni and @Harvard. We suggest the following hashtags: #CarbonNeutral, #RenewableEnergy, and #China.
- Some responses to the submission information requirements in the decision emails such as the Feature image issues are provided at the end of this cover letter.

Please see the revised manuscripts, the revised author checklist and the following responses for details.

Response to the Reviewers

Reviewer #1

Comment 1: I thank the authors for their careful and complete response to my comments. Overall, the authors are to be congratulated for the quality of the work. The considerable amount of efforts is tangible, and has led to a high-quality paper. I have no further comments.

Reply: Thank you for providing valuable comments and acknowledging the contributions of our work. This paper provided quantified conclusions and policy implications on carbon neutrality in China's power system and established the dataset for future work about China's energy transition. We hope that the data and methods presented in this paper can provide basic tools for future research about the carbon-neutral transition of China's energy system.